# Field Theoretical Approach for Signal Detection in Nearly Continuous Positive Spectra I: Matricial Data

**DOI:** 10.3390/e23091132

**Published:** 2021-08-31

**Authors:** Vincent Lahoche, Dine Ousmane Samary, Mohamed Tamaazousti

**Affiliations:** 1Université Paris-Saclay, CEA, List, F-91120 Palaiseau, France; vincent.lahoche@cea.fr (V.L.); mohamed.tamaazousti@cea.fr (M.T.); 2International Chair in Mathematical Physics and Applications (ICMPA-UNESCO Chair), University of Abomey-Calavi, Cotonou 072 P.O. Box 50, Benin

**Keywords:** renormalization group, field theory, phase transition, big data, principal component analysis, signal detection, information theory

## Abstract

Renormalization group techniques are widely used in modern physics to describe the relevant low energy aspects of systems involving a large number of degrees of freedom. Those techniques are thus expected to be a powerful tool to address open issues in data analysis when datasets are highly correlated. Signal detection and recognition for a covariance matrix having a nearly continuous spectra is currently one of these opened issues. First, investigations in this direction have been proposed in recent investigations from an analogy between coarse-graining and principal component analysis (PCA), regarding separation of sampling noise modes as a UV cut-off for small eigenvalues of the covariance matrix. The field theoretical framework proposed in this paper is a synthesis of these complementary point of views, aiming to be a general and operational framework, both for theoretical investigations and for experimental detection. Our investigations focus on signal detection. They exhibit numerical investigations in favor of a connection between symmetry breaking and the existence of an intrinsic detection threshold.

## 1. Introduction

Statistical physics was born to deal with systems involving a very large number of interacting degrees of freedom, to extract relevant features at large scales, when classical mechanics are no longer a feasible option [1]. These relevant features generally take the form of an effective description involving a small number of parameters related to very large number of parameters used to describe microscopic states. From the point of view of information theory, statistical physics looks like a consequence of a statistical inference based on the maximum entropy estimate, disregarding the specific aspects of the microscopic problem, and it is for this reason a general paradigm [2,3,4]. In the last century, Wilson and Kadanoff [5,6,7,8,9] discovered that an effective description valid for large scales can be deduced from the knowledge of a microscopic description following a coarse-graining procedure called renormalization group (RG), which averages recursively over the fluctuations that have short wavelengths, with respect to a reference scale. In modern physics [10,11], RG becomes a powerful tool to address the questions of universality and simplicity of the large-scale physical laws [12,13,14,15,16,17,18,19].

Modern data analysis aims to deal with very large datasets, which are strongly correlated, and principal component analysis (PCA) is one of the most popular methods to extract information, i.e., to find relevant features [20,21,22,23,24,25,26,27,28,29,30,31]. Even though different realisations are present in the literature, the principle is always the same. PCA is a linear projection onto the lower dimensional subspace spanned by the eigendirections corresponding to the larger eigenvalues (the relevant features). For the datasets taking the form of a suitably mean-shifted and normalized N×P matrix $X_{ai}$, with a∈{1,⋯,P} and i∈{1,⋯,N}, the covariance matrix C (if one wishes to work with different random variables, for instance associated to different systems, one should also reduce the matrix *X*, and work with the correlation matrix) is defined as the average of $X^{T}X$, describing correlations between type-*i* variables. Standard PCA works well when the largest eigenvalues can be easily distinguished from the other ones. In this case, a small number of modes captures the most relevant features of the covariance. Such an effective description is reminiscent of the famous *large river effect* of the RG flow in statistical physics, referring to the general property of this flow to be dragged toward a finite-dimensional subspace corresponding to relevant and marginal operators for sufficiently large scales [32].

The connection between PCA and RG can be traced from information theory [33,34,35,36,37,38,39,40] as a consequence of the ability of the RG to extract large scale relevant features of a microscopic system. From an information theory point of view, the RG describes how a theory valid for small distance physics flows toward a simpler theory (i.e., with a reduced set of parameters) at larger distance, as information is progressively lost (due to coarse-graining). In turn, the inference problem of recovering the elementary theory from the knowledge of the large scale simpler model is the same as finding the equivalent class of elementary models having the same large scale limits. The distinction between elements of the different equivalent family is based on the existence of an intrinsic criterion which quantifies the intrinsic ability for a perturbation to a given microscopic state to survive at large scale. The relevant operators of quantum fields, that survives at a large distance, are the only ones whose provide a clean distinction between asymptotic states for some large scale observer. Relevance can be defined intrinsically from information theory, according to a specific notions of distance between states. In information geometry theory, where the state space looks as a differentiable manifold, a computable notion of distance is given by the Fisher information metric, and the notion of equivalence class can be defined with respect to this distance, since all the asymptotic states having distance smaller than some working precision are undistinguishable [36,37,38,39].

In the case of a continuous spectrum [21,22,41,42], the standard PCA fails to provide a clean separation between noise and information. We propose to exploit the link between PCA and RG to address this separation with an objective physical criterion. A first step in this direction was done in [20,21,22]. The authors considered an effective field theory framework, where the separation between information and noise looks like an arbitrary cut-off scale Λ in the eigenvalue spectrum of the covariance matrix (see Figure 1). In [23], the authors introduced the non-perturbative Wetterich–Morris framework. In that formalism the bare (i.e., the microscopic) action is left unchanged, but infrared contributions are suppressed from the effective action. Hence, the formalism focus on the effective action $\Gamma_{k}$ for the integrated-out degrees of freedom up to the scale *k* rather than the microscopic action for the remaining (infrared) degrees of freedom. In some sense, we rather focus on determining what the “noise” is than what the “information” is, and *k* looks as an IR rather UV cut-off (the notion of infrared (IR) and ultraviolet (UV) are related, respectively, to the smaller and higher values of the scale parameter *k*.). In this previous study however, the investigations of the author were essentially restricted to the power-counting aspects for distributions around the universal Marchenko–Pastur (MP) noise [43]. The conclusions were: (1) for a purely noisy data the first quartic perturbation to the Gaussian distribution is relevant from coarse-graining; (2) a strong enough signal must change the power counting to make the perturbation irrelevant, and the effective description goes perturbatively toward the Gaussian point.

The aim of the following work is to consider the same question and to go beyond these dimensional aspects. More precisely, we address the following question: within our field theoretical formalism, is it possible to find objective criteria to decide if a continuous spectrum associated to a dataset contains information or not? Focusing on deformations around MP law, our result shows that in this formalism, the presence of a signal in the spectrum can be identified from a symmetry breaking corresponding to a non-zero value of the field theory vacuum. The symmetry is progressively restored as the signal strength is turned to zero. Note that the conclusions of this paper have been extended to tensorial PCA in [25].

Let us note that phase transitions are usually associated to signal detection in PCA [44,45,46]. A classic example is provided by the one-spiked matrix model [47], which exhibits a sharp phase transition in the larger eigenvalue distribution. In that elementary case the signal is materialized by a single vector $u\in R^{N}$ with length 1 and the noise by a N×N matrix *M* along the Gaussian orthogonal ensemble. Let us denote by λ the size (intensity) of the signal. For λ<1, the largest eigenvalue $\lambda_{c}$ in the sum $M+\lambda uu^{T}$ follows semi-circle law with Tracy–Widom distribution for N→∞. In contrast, for λ>1, the largest eigenvalue converges toward $\lambda_{c}=\lambda +1/\lambda$, and its statistic follows a Gaussian error function. From the point of view of signal detection, this result as a consequence that as soon as λ>1, the signal can be easily detected using standard iterative methods; whereas detection is impossible in practice for λ<1. The main difference with our point of view is that we do not focus on the largest eigenvalue distribution, but on the ability of the spectrum to support large-scale non-trivial structures embedded in a field theory.

The manuscript is organized as follows. Section 2 provides a summary of the field theoretical framework introduced in [23], specifying some subtle points not discussed in this previous work, especially with regard to the kinetic classical action. Moreover, anticipating the results of the next section, we discuss the relevance of interactions and argue the existence of a “wall” in generalized momenta, defining a physical cut-off, below which the relevant sector diverges and the field theoretical embedding breaks down. Working above this wall, we show that only a few number of local couplings are relevant, essentially quartic and sixtic couplings for small perturbations around MP law. We close the section with the Wetterich–Morris non-perturbative formalism [21,22,32,33,34,35,36,37,38,39,40,41,42,43,44,45,46,47,48,49,50,51,52], that we discuss in this context. Section 3 is devoted to an analysis of the exact RG equation using standard local potential approximation (LPA) and its improved version (LPA${}^{\prime }$) which takes into account field renormalization effects and anomalous dimension. Within this approximation scheme, we are able to identify the presence of a signal (materialized by a few number of discrete spikes disturbing the purely noisy data) as a lack of symmetry restoration in the deep IR, for k∼1/N. Finally, in Section 4, we summarize our investigations, and discuss some open issues, together with a plan to address them in the future.

## 2. The RG in Field Theory

Despite its origins in particle physics, the RG is probably one of the most important and universal concepts discovered during the last century. RG enables to understand physics at different scales. Technically, in the field theoretical framework, this is a consequence of the ability of the microscopic degrees of freedom to be reabsorbed by a set of parameters, that predictivity is required to be finite to design an effective field theory. Such a theory has, thus, the property to be valid only up to a certain scale, where the more fundamental degrees of freedom cannot be distinguished from their effective description. The same procedure can then be repeated, resulting in an effective chain of theories. This *coarse-graining* is at the core of the RG idea, as originally formulated by Kadanoff and Wilson [5,6,7,8,9] (see Figure 2). In particular, the RG aims to address the following question: to what extent can two different microscopic states be distinguishable under coarse-graining? To be more formal, let us consider a system built with a large number *N* of interacting degrees of freedom $\zeta_{i}$ for i∈[1,N]. A microscopic state is a set $\zeta \equiv \{\zeta_{1},\zeta_{2},\cdots ,\zeta_{N}\}$. The nature of the elementary states $\zeta_{i}$ describing a single degree of freedom depends on the problem that one considers (for the standard Ising model, for instance [53], $\zeta_{i}$’s are discrete variables $\zeta_{i}=\pm 1$). Assuming the maximum entropy prescription, these states are associated to a probability distribution $p\left[\zeta \right]=e^{-S[\zeta ]}$, where S[ζ] is called *classical action* or *fundamental Hamiltonian* in physics. This microscopic level is conventionally called *ultraviolet scale* (UV scale), and the dominant configurations, say classics, are given by the saddle point equation $\partial S/\partial \zeta_{i}=0$. The momenta of the distribution are generated by:(1)$$Z\left[j\right]=\sum_{\zeta }\;p\left[\zeta \right]e^{j\zeta }\;,$$
where $j\equiv \{j_{1},j_{2},\cdots ,j_{N}\}$ and $j\zeta :=j_{1}\zeta_{1}+\cdots +j_{N}\zeta_{N}$. Moreover, the formal sum ∑ have to be replaced by a functional path integral for continuous degrees of freedom. The *classical field*
$M:=\{M_{i}\}$ is defined as the means value of $\zeta_{i}$: $M_{i}:=\sum_{\zeta }\;\zeta_{i}\;p\left[\zeta \right]$. The cumulants of the distribution are generated by the *free energy*
W[j]=lnZ[j], taking successive derivatives with respect to the source *j* and setting j=0. In the field theoretical vocabulary, W[j] is the generating functional of connected correlations functions (i.e., the correlations functions which cannot be factorized as a product of two-point correlation functions). The physical configurations for *M* are fixed, for j=0 by an equation taking the same form as the saddle point equation for ζ, but involving *the effective action*
Γ[M], $\partial \Gamma /\partial M_{i}=0$. This effective action is formally defined as the Legendre transform of the free energy:(2)Γ[M]+W[j]=jM. We call *infrared scale* (IR scale) the effective description where fluctuations are integrated out. From a statistical point of view, Γ[M] is the generating function of one particle irreducible (1PI) diagrams or effective vertices, in the sense that they represent effective couplings between components of the field *M* entering in the construction of the functional Γ[M].

Then, the Wilson RG procedure [17,18] assumes a progressive dilution of the information, integrating-out progressively the fluctuating degrees of freedom following a given slicing. In that way, RG constructs a path from UV to IR scales (see Figure 2), in which each step provides an effective description, associated to an effective classical action describing fluctuations of non-integrated degrees of freedom. Thus, the effect of the microscopic degrees of freedom that we ignore is hidden in the parameters defining this action. Thus, RG transformations define a mapping from an action to another at different scales, and the successive positions of the classical action through the functional space of allowed actions construct a trajectory, starting in the UV and ending in the IR. This functional space is usually called *theory space*. Along this trajectory, the couplings (i.e., the parameters defining the action) change; and the RG equations take the form of a dynamical system describing this running behavior of the couplings along RG trajectories.

However, the existence of such a path is guaranteed only if it is possible to define a criterion for the choice of the renormalization scale of such fluctuations. In standard field theory, this criterion is given by the energy of the modes, the high energy modes being associated to small scales whereas low energy modes are associated to large scales. These energy levels correspond to the eigenvalues of some physically relevant operator. In standard field theory, for instance, for a classical action describing a scalar field ϕ(x) on $R^{d}$,
(3)$$S\left[\phi \right]:=\frac{1}{2}\;\int_{R^{d}}d^{d}x\;\phi \left(x\right)\left(-\Delta +m^{2}\right)\phi \left(x\right)+\frac{g}{4!}\;\int_{R^{d}}d^{d}x\;\phi^{4}\left(x\right)\;,$$
the operator allowing to classify the modes is the kinetic operator $K=\Delta +m^{2}$; or simply the Laplacian Δ, whose eigenvectors are the Fourier modes.

### 2.1. A Field Theoretical Embedding for Data Analysis

As in statistical physics, in the big data area, a state is a point in a space with a very large number of dimensions. PCA, in turn, aims to find the most relevant features out of a very large number of variables. In the case of a continuous spectrum, the relevance is fixed by some sensitivity threshold, discriminating between large eigenvalues that we keep and small eigenvalues that we discard. In other words, PCA constructs effective descriptions, which are valid as long as we can ignore the small eigenvalue effects. This picture is reminiscent of what RG do. From this, it could exist a criterion to distinguish between a noisy spectrum and another containing information, based on the differences between their respective effective asymptotic states. Let us consider the example of the field theory described by the classical action (Equation 3). The dimension of coupling constants like *g* depends on the dimension of the background space $R^{d}$. In this example, [g]=d−4. The relevance in the vicinity of the Gaussian fixed point (g≈0) depends on the value of this dimension. For d>4, the operator $\phi^{4}\left(x\right)$ is irrelevant, meaning that for sufficiently large scale, the theory is essentially Gaussian. In the opposite situation, for d<4, the RG flow moves away from the Gaussian fixed point (The point $m^{2}=g=0$ is a fixed point of the RG flow, thus any partial integration leads to another Gaussian model in virtue of the Gaussian integration properties. Thus, the relevance of the operators in the deep IR is usually determined by the dimension of the space.) However, the latter determines also the shape of the distribution for the Laplacian eigenvalues $p^{2}$, which is $\rho \left(p^{2}\right)={\left(p^{2}\right)}^{d/2-1}$, and relevance can be alternatively viewed as a property of the momentum distribution, without reference to the background space dimension. This is exactly what the authors of references [20,21,22,23] did: the scaling dimension is defined through the coarse graining, from the requirement that no explicit scale dependence occurs in the flow equations, excepts eventually at the linear level (see Section 3.1).

We propose a framework allowing to construct a field theoretical approximation of the fundamental RG flow. This point of view is familiar in condensed matter physics, and especially in the physics of critical phenomena. The classical example is the Ising model, whose critical behavior may be well approximated by an effective field theory in the critical domain [7]. Let us consider a set of *N* random variables, $\phi =\left\{\phi_{1},\phi_{2},\cdots ,\phi_{N}\right\}\in R^{N}$; providing an archetypal example of field. Moreover, we assume that there exists a distribution p[ϕ] able to reproduce the covariance matrix C, at least for sufficiently large scale (in eigenvalue space). An elementary formal realization of this is given by the Gaussian states:(4)$$p\left[\phi \right]\propto \exp\left(-\frac{1}{2}\sum_{i,j}\;\phi_{i}C_{ij}^{-1}\phi_{j}\right)\;,$$
ensuring that $\langle \phi_{i}\phi_{j}\rangle =C_{ij}$. The bracket notation 〈X[ϕ]〉 defines the mean value of *X* with respect to the distribution p[ϕ]. For such a Gaussian description, all the non-vanishing momenta of the distributions, $\langle \phi_{i}\phi_{j}\phi_{k}\cdots \rangle$ reduce to a sum of the product of 2-points function following Wick theorem, and only the second cumulant does not vanish. In the field theory language, a theory with this property is said to be *free*. From an RG point of view, this description makes sense only if the Gaussian point is stable, i.e., if any perturbation around the Gaussian point ends up disappearing after some steps of the RG. In the next section we show that for MP law the Gaussian point is unstable. Moreover for a realistic dataset, correlations for *n*-point functions fail to be a product of 2-point functions, as a purely Gaussian law would have required. In the field theory language, this failing of the Gaussian description indicates the presence of *interactions*, materialized as non-Gaussian terms in the action. The coupling $g\int \phi^{4}\left(x\right)$ in (Equation 3) provides an elementary example.

We, therefore, consider an interacting field theory. In standard field theory, there exist powerful principles, inherited from physics or mathematical consistency, to guide the choice of interactions, and the relevant domains of the theory space. In the absence of such a guide, we use the same simplicity argument already considered in [20,23]. We focus on purely local interactions of the form $g\sum_{i}\phi_{i}^{2n}$, with fields interacting on the same point, with the same coupling constant. In that way, near to the Gaussian point our distribution p[ϕ] is suitably expanded as
(5)$$p\left[\phi \right]\propto \exp\left(-\frac{1}{2}\sum_{i,j}\;\phi_{i}{\tilde{C}}_{ij}^{-1}\phi_{j}-\frac{g}{4!}\sum_{i}\;\phi_{i}^{4}+\cdots \right)\;.$$
Note that, in our assumptions we kept only even interaction terms, ignoring for instance couplings like $\phi_{i}^{3}$. This hypothesis is equivalent to assume the reflection symmetry (note that, truncating around quartic interactions, adding a term like $\sum_{i,j}\phi_{i}^{2}\phi_{j}^{2}$, which is invariant under to the rotational group O(N), enlarges the discrete group $Z_{2}$ to the hypercubic symmetry. Thus, the purely local model is, with this respect, the less structured one.) $\phi_{i}\to -\phi_{i}$. Finally, note that, in principle, ${\tilde{C}}^{-1}\neq C^{-1}$. Indeed, what is known “empirically” is the full 2-point function $C_{ij}$; and the probability distribution must be:(6)$$\int d\phi \;\phi_{i}\phi_{j}p\left[\phi \right]=C_{ij}\;.$$
and from perturbation theory
(7)$$\int d\phi \;\phi_{i}\phi_{j}p\left[\phi \right]={\tilde{C}}_{ij}+O\left(g\right)\;.$$
Note that, from an information theory point of view, probability density (Equation 5) corresponds to the maximum entropy solution, compatible with constraint (Equation 5) and the existence of undetermined non-Gaussian correlations. From that point of view, the model is minimal in the sense that it carries the least possible structure, as stated in [20]. In that setting $C_{ij}={\tilde{C}}_{ij}$ only at first order, and when non-Gaussian contributions cannot be neglected, ${\tilde{C}}_{ij}$ receives quantum corrections, depending on the couplings in a non-trivial way. Inferring the Gaussian kernel ${\tilde{C}}_{ij}$ from the knowledge of $C_{ij}$ is a very challenging problem in field theory. In some approximation schemes however, relevant to extracting non-perturbative information about the behavior of the RG flow, this difficulty is not a limitation of our investigation. In the local potential approximation that we will consider in this paper for instance, we assume that $C_{ij}^{-1}$ and ${\tilde{C}}_{ij}^{-1}$ differ by a constant, $C_{ij}^{-1}={\tilde{C}}_{ij}^{-1}+k$; the constant *k* capturing all the quantum corrections. One would expect that an approximation works, essentially, in the region of small eigenvalues for $C_{ij}^{-1}$, the IR regime in the field theoretical language. We hope that our methods can track the presence of a signal. We will return on this discussion in Section 3.1. A finer analysis would require more elaborate methods, beyond the scope of this paper. We are then able to construct an approximation of the RG flow, which is not autonomous due to the lack of dilatation invariance of the eigenvalue distribution for $C_{ij}^{-1}$ (see Section 3). Finally, let us mention a remark about the field theoretical embedding. In Section 3, we show that even for the MP law, the number of relevant interaction becomes arbitrarily large in the first 66% of the smallest eigenvalues. This introduces an unconventional difficulty in field theory, which can be alternatively viewed as a limitation of the field theory approximation. The breaking down of the field theory up to a certain scale is not a novelty. It is well known for instance that the Ising model behaves like an effective $\phi^{4}$ field theory like (Equation 3) in the vicinity of the ferromagnetic transition. Thus, a failure of the field theoretical approximation may be alternatively viewed as a signal that a more elementary description is required. Then, it is interesting to remark that the field theory considered in (Equation 5) may be essentially deduced from the Ising-like model:(8)$$p_{\mathrm{Ising}}\left(\left\{S\right\}\right)\propto \exp\left(\frac{1}{2}\;S_{i}C_{ij}S_{j}\right)\;,$$
where the $S_{i}=\pm 1$ are discrete Ising spins. Indeed, introducing *N* reals variables $\phi_{i}$, and using the standard Gaussian trick to rewrite the quadratic term in $S_{i}$;
(9)$$p_{\mathrm{Ising}}\left(\left\{S\right\}\right)\propto \int \prod_{i}d\phi_{i}\;\exp\left(-\frac{1}{2}\;\phi_{i}C_{ij}^{-1}\phi_{j}+S_{i}\phi_{i}\right)\;.$$ Thus, summing over $\left\{S_{i}\right\}$ configurations generates an effective $\sum_{i}\cosh\left(\phi_{i}\right)$; and expanding it in power of $\phi_{i}$ reproduces the terms appearing in the local expansion in (Equation 5). The model described by (Equation 8) is reminiscent of the standard spin-glass models, as the Sherrington–Kirkpatrick model [54,55,56,57]. Its derivation, moreover, follows directly from the maximum entropy prescription with constraint (Equation 6) if we assume to work with discrete spins. We can alternatively see our model of field theory as coming from this binary model, derived from a principle of maximum entropy with fixed correlations [2,3,4].

### 2.2. The Model

In this section, we provide a mathematical definition of the field theoretical model that we consider, an extended discussion can be found in [20,23]. Following these references, we work in the eigenbasis of the matrix $C_{ij}^{-1}$, more suitable for a coarse-graining approach. Note that, with our assumption, this eigenbasis is the same as the one for ${\tilde{C}}_{ij}^{-1}$. In that way, the Gaussian (or kinetic) part of the classical action of p[ϕ] takes the form;
(10)$$S_{\mathrm{kinetic}}\left[\psi \right]=\frac{1}{2}\sum_{\mu }\psi_{\mu }\lambda_{\mu }\psi_{\mu }\;,$$
where $\lambda_{\mu }$ denote the eigenvalues of ${\tilde{C}}_{ij}^{-1}$, labeled with the discrete index μ; and the fields $\left\{\psi_{\mu }\right\}$ are the eigen-components of the expansion of $\phi_{i}$ along the normalized eigenbasis $u_{i}^{\left(\mu \right)}$:(11)$$\psi_{\mu }=\sum_{i}\phi_{i}u_{i}^{\left(\mu \right)}\;,\;\sum_{j}{\tilde{C}}_{ij}^{-1}u_{j}^{\left(\mu \right)}=\lambda_{\mu }u_{i}^{\left(\mu \right)}\;.$$ Let then $m^{2}$ be the smallest eigenvalue. We introduce the positive definite quantities $p_{\mu }^{2}:=\lambda_{\mu }-m^{2}$. This way, the kinetic action takes the form of a standard kinetic action in field theory:(12)$$S_{\mathrm{kinetic}}\left[\psi \right]=\frac{1}{2}\sum_{\mu }\psi_{\mu }\left(p_{\mu }^{2}+m^{2}\right)\psi_{\mu }\;.$$ In the continuum limit, for *N* sufficiently large, it is suitable to use the empirical eigenvalue distribution χ(λ) and to replace sums by integrals. This distribution is empirically inferred from $\sum_{\mu }\delta \left(\lambda -\lambda_{\mu }\right)/N$, and the distribution $\rho (p_{\mu }^{2})$ for the *momenta*
$p_{\mu }^{2}$ follows. Moreover, with our assumptions on ${\tilde{C}}_{ij}^{-1}$, this distribution can be directly deduced from the spectrum of $C_{ij}^{-1}$. Finally, for purely random matrices with i.i.d entries, the MP theorem states that the empirical distribution has to converge toward an analytic form that can be used to do exact computations.

It is, however, difficult to deal with interactions in this formalism. A simplification, already considered by the authors of [23] is to work with momentum-dependent field ψ(p) rather than with the field $\psi_{\mu }$ (labeled with the positive quantity $p_{\mu }^{2}$). This can be motivated by the observation that the model (Equation 5) is no rather fundamental than any other model able to reproduce effective correlations in some approximation scheme. In local approximation, that we consider in this paper, the two formulations are expected to be in the same universality class when we define locality in momentum space as:

**Definition** **1.***An interaction is said to be local of order P if it involves P fields and if it is conservative, i.e., if it is of the form:*(13)$$U\left[\psi \right]\propto \sum_{\left\{p_{\alpha }\right\}}\;\delta_{0,\sum_{\alpha =1}^{P}p_{\alpha }}\;\prod_{\alpha =1}^{P}\;\psi \left(p_{\alpha }\right)\;.$$*where $\delta_{p,p^{\prime }}$ denotes the standard Kronecker delta, equal to* 1 *for $p=p^{\prime }$ and zero otherwise. By extension, we say that a functional U[ψ] is local if its expansion in power of ψ involves only local terms.*

This definition follows the standard one in classical and quantum field theory, and it is moreover consistent with the idea that locality is defined ‘‘at contact point” in the original representation (Equation 5).

### 2.3. Wetterich–Morris Framework

Among the different incarnations of the Kadanoff–Wilson’s coarse-graining idea, the Wetterich–Morris (WM) framework has the advantage to be well suited to non-perturbative approximation methods [58,59]. Rather than Kadanoff–Wilson approach, which focuses on the effective classical action $S_{k}$ for IR modes below the scale *k*, the WM formalism focuses on the effective averaged action $\Gamma_{k}$; i.e., the effective action for integrated-out modes above the scale *k*. As recalled in the previous section, the fundamental ingredient to describe IR scales, when all degrees of freedom have been integrated out is the effective action Γ[M], which is defined as the Legendre transform of the free energy W[j] (Equation (Equation 2)); the classical field $M=\{M_{\mu }\}$ being defined as:(14)$$M_{\mu }=\frac{\partial W[j]}{\partial j_{\mu }}\;.$$ The starting point of the WM formalism is to modify the classical action S[ψ] adding a scale dependents term $\Delta S_{k}\left[\psi \right]$, depending on a continuous index *k* running from k=Λ, for some fundamental UV scale Λ, to k=0. In such a way, we define a continuous family of models, described by a free energy $W_{k}\left[j\right]$ defined as:(15)$$W_{k}\left[j\right]:=\ln\int \left[d\psi \right]p\left[\psi \right]e^{-\Delta S_{k}\left[\psi \right]+\sum_{\mu }j\left(p_{\mu }\right)\psi \left(p_{\mu }\right)}\;.$$ The regulator function $\Delta S_{k}\left[\psi \right]$ behaves like a mass, whose value depends on the momentum scale:(16)$$\Delta S_{k}\left[\psi \right]=\frac{1}{2}\sum_{\mu }\;\psi \left(p_{\mu }\right)r_{k}\left(p_{\mu }^{2}\right)\psi \left(-p_{\mu }\right)\;.$$ The momenta scale $r_{k}\left(p_{\mu }^{2}\right)$ provides an operational description of the Kadanoff–Wilson’s coarse-graining procedure, and it is chosen, such that:$r_{k=0}\left(p^{2}\right)=0\;\;\forall p^{2}$, meaning that for k=0, $W_{k}\equiv W$, all the fluctuations are integrated out;$r_{k=\Lambda }\left(p^{2}\right)\gg 1$, meaning that in the deep UV, all fluctuations are frozen with a very large mass;$r_{k}\left(p^{2}\right)\approx 0$ for $p^{2}/k^{2}<1$, meaning that high energy modes with respect to the scale $k^{2}$ are essentially unaffected by the regulator. In contrast, low energy modes must have a large mass which decouples them from long-distance physics.

The two boundaries conditions ensure that we recover the effective descriptions, respectively, in the UV limit, where no fluctuations are integrated out, and in the deep IR where all the fluctuations are integrated out. In other words, we interpolate between the classical action S and the effective action Γ. This can be achieved by introducing the effective averaged action $\Gamma_{k}$ defined as:(17)$$\begin{array}{cc} \Gamma_{k}\left[M\right]+W_{k}\left[j\right] & =\sum_{\mu }j\left(p_{\mu }\right)M\left(p_{\mu }\right)-\frac{1}{2}\sum_{\mu }\;M\left(p_{\mu }\right)r_{k}\left(p_{\mu }^{2}\right)M\left(-p_{\mu }\right)\;, \end{array}$$
such that $\Gamma_{k=0}\equiv \Gamma$ and, from the conditions on $r_{k}$, $\Gamma_{k=\Lambda }∼S$. The meaning of $\Gamma_{k}$ is illustrated in the Figure 3. Along the path from k=Λ to k=0, $\Gamma_{k}$ goes through the theory space, and the different coupling changes. The dynamics of the couplings can be deduced considering a small variation k→k+dk, and we can show that $\Gamma_{k}$ obeys to the WM equation [58,59]:(18)$${\dot{\Gamma }}_{k}=\frac{1}{2}\;\sum_{\mu }{\dot{r}}_{k}\left(p_{\mu }^{2}\right){\left(\Gamma_{k}^{\left(2\right)}+r_{k}\right)}_{\mu ,-\mu }^{-1}\;.$$ This equation is the one that we will use in this paper to investigate RG flow for datasets. The dot notation ${\dot{\Gamma }}_{k}$ represents the partial derivation of $\Gamma_{k}$ with respect to the scale *k*.

## 3. RG, from Theory to Numerical Investigations

In this section, we investigate the behavior of the RG flow, focusing on the evolution of the field expectation value and symmetry restoration aspects. However, since the functional space has infinite dimensions, solving the non-perturbative Equation (Equation 18) is a difficult task, requiring approximations that we discuss as well.

As a first approximation, we focus on the symmetric phase [60,61,62,63,64,65,66], which can be defined as the region of the whole phase space where it makes sense to expands the averaged effective action $\Gamma_{k}\left[M\right]$ in the power of *M*, around M=0. Regions where M=0 become unstable vacua and the field expansion can be improved by an expansion around a non-zero vacuum. This works well in the local potential approximation (LPA), neglecting the momentum dependence of the classical field. Corrections to the strict LPA take the form of a perturbative expansion in the power of $p^{2}$, called *derivative expansion* (this terminology is inherited from the standard field theory, where an expansion in the power of the momentum $p^{2}$ is nothing but an expansion in the power of Δ, the standard Laplacian in $R^{d}$.) (DE). In this paper, we consider only the first terms in the derivative expansion, provided by the kinetic action contribution $\int \frac{1}{2}p^{2}M\left(p\right)M\left(-p\right)$ to $\Gamma_{k}\left[M\right]$. In strict LPA, the coefficient in front of $p^{2}$ (the field strength) remains equal to 1. A slight improvement to the LPA, called LPA${}^{\prime }$ takes into account the field strength flow Z(k): $\int \frac{1}{2}p^{2}M\left(p\right)M\left(-p\right)\to \int \frac{1}{2}Z\left(k\right)p^{2}M\left(p\right)M\left(-p\right)$, so that the anomalous dimension does not vanish. We will consider both these approximations, showing explicitly that the corrections provided to LPA${}^{\prime }$ remain small into the range of scales that we consider, thus ensuring the validity of the LPA, as well the reliability of our conclusions.

### 3.1. Solving the Exact RG Equation into the Symmetric Phase

#### 3.1.1. Generalities

As explained before, a truncation is generally required to solve the RG Equation (Equation 18). In other words, a truncation is nothing but an ansatz for $\Gamma_{k}$, and, thus, a specific parametrization of a finite-dimensional region of the full phase space. The reliability of the method is, however, no guarantee in general, and a deep inspection is always required to validate the conclusions of the truncations. Generally, there are two main sources of shortcomings.

The first one comes from the choice of the regulator $r_{k}$. Indeed, formally, the boundary conditions ensured by $r_{k}$ and $\Gamma_{k}$ are such that different choices for $r_{k}$ lead to different trajectories into the theory space, with the same boundary conditions $\Gamma_{k=0}=\Gamma$. This formal device however does not survive the truncation procedure in general, and it is well known that a spurious dependence on the regulator appears for physically relevant quantities like critical exponents. The knowledge of exact results or exact relations enables in favorable cases, to improve the choice of the regulator. Some general considerations based on optimization criteria can be of some help in other cases [67,68,69]. For our purpose, since we essentially focus on the shape of the effective potential rather than on the specific value of a physical quantity, one expects that such dependence is not too relevant.

The second one is the choice of truncation. A general criterion is based on the relative relevance of the different ingredients entering in the definition of $\Gamma_{k}$. In the worst case, the parametrization may conflict with exact relations, coming, for instance, from symmetries like Ward identities [60,61,62,63,64,65,66]. Once again, one expects that such a pathological effect is not likely to appear here.

In this section, we aim to focus on the symmetric phase, where $\Gamma_{k}$ is assumed to be well expanded in the power of *M*. With this assumption, it is suitable to write $\Gamma_{k}\left[M\right]=\Gamma_{k,\mathrm{kin}}\left[M\right]+U_{k}\left[M\right]$, where $\Gamma_{k,\mathrm{kin}}\left[M\right]$, the kinetic part, keeps only the quadratic terms in *M* and $U_{k}\left[M\right]$, the potential, is expanded in power of *M* higher than 2. In the LPA, the potential $U_{k}\left[M\right]$ is a purely local function, in the sense of the Definition 1. Moreover, we assume that $U_{k}$ is an even function, i.e., that the symmetry M→−M holds. In contrast, $\Gamma_{k,\mathrm{kin}}\left[M\right]$, whose inverse propagates the local modes, may involve non-local contributions, and its generic parametrization reads as:(19)$$\Gamma_{k,\mathrm{kin}}\left[M\right]=\frac{1}{2}\;\sum_{p}\;M\left(-p\right)\left(Z\left(k,p^{2}\right)p^{2}+u_{2}\left(k\right)\right)M\left(p\right)\;,$$
where $Z(k,p^{2})$ expands in power of $p^{2}$ as $Z\left(k,p^{2}\right)=Z\left(k\right)+O\left(p^{2}\right)$. In this paper, we focus on the first order of the DE, keeping only the term of order ${\left(p^{2}\right)}^{0}$ in the expansion of $Z(k,p^{2})$. In the symmetric phase moreover, assuming that $U_{k}\left[x\right]$ is an even function, the flow equation for Z(k) vanishes exactly. Thus, it is suitable to fix the normalization of fields, such that Z(k)=1∀k.

As explained in Section 2.1, the field theory framework that we consider is non-conventional in the sense that the full kinetic action is known in the deep IR, but not at the microscopic level. We, thus, have to infer the microscopic kinetic action from the IR regime. Such an inference problem is reputed to be a hard problem (Figure 4), especially because the coarse-graining is not invertible. A consequence of this is the large river effect [70]. Usually, when a sufficiently large number of degrees of freedom have been integrated out, all the RG trajectories converge toward a finite-dimensional region of the full phase space, spanned by relevant and marginal (by power counting) interactions. In other words, different microscopic physics may have the same effective behavior at sufficiently large scale, the difference, spanned by irrelevant (i.e., non-renormalizable) interactions falling below the experimental precision threshold on a large enough scale. Thus, the best compromise is an equivalence class of microscopic models, that are not distinguishable (up to some experimental precision) in the deep IR. This hard inference problem is simplified within the LPA, because the expression for the effective kinetic action differs only by the mass parameter $u_{2}\left(k\right)$.

The derivation of the flow equations follows the general strategy [65]. Taking the second derivative of the Equation (Equation 18) with respect to $M_{\mu }$, we get:(20)$${\dot{\Gamma }}_{k,\mu_{1}\mu_{2}}^{\left(2\right)}=-\frac{1}{2}\sum_{\mu }{\dot{r}}_{k}\left(p_{\mu }^{2}\right)G_{k,\mu \mu^{\prime }}\Gamma_{k,\mu^{\prime }\mu^{\prime \prime }\mu_{1}\mu_{2}}^{\left(4\right)}G_{k,\mu^{\prime \prime }\mu }\;.$$ The different terms involved in this expression can be explicitly derived from the truncation. Indeed, from:(21)$$\begin{array}{cc} \Gamma_{k}\left[M\right] & =\frac{1}{2}\sum_{p}M\left(-p\right)\left(p^{2}+u_{2}\left(k\right)\right)M\left(p\right)\\ \; & +\frac{u_{4}\left(k\right)}{4!N}\;\sum_{\left\{p_{i}\right\}}\;\delta \left(\sum_{i}p_{i}\right)\prod_{i=1}^{4}M\left(p_{i}\right)\\ \; & +\frac{u_{6}\left(k\right)}{6!N^{2}}\;\sum_{\left\{p_{i}\right\}}\delta \left(\sum_{i}p_{i}\right)\prod_{i=1}^{6}M\left(p_{i}\right)+O\left(M^{6}\right)\;, \end{array}$$
we straightforwardly deduce that:(22)$$\Gamma_{k,\mu_{1}\mu_{2}}^{\left(2\right)}=\delta_{p_{\mu_{1}},-p_{\mu_{2}}}\left(p_{\mu_{1}}^{2}+u_{2}\left(k\right)\right)\;,$$
and:(23)$$\Gamma_{k,\mu_{1}\mu_{2}\mu_{3}\mu_{4}}^{\left(4\right)}=\frac{g}{4!N}\sum_{\pi }\delta_{0,p_{\pi (\mu_{1})}+p_{\pi (\mu_{2})}+p_{\pi (\mu_{3})}+p_{\pi (\mu_{4})}}\;,$$
where π denotes elements of the permutation group of four elements. Note that, the origin of the factors 1/N and $1/N^{2}$ can be easily traced. As we will see below, the 1/N in front of $u_{4}$ ensures that (Equation 20) can be rewritten as an integral in the large *N* limit, involving the effective distribution $\rho (p^{2})$. The $1/N^{2}$ in front of $u_{6}$ ensures that all the contributions to the flow of $u_{4}$ receive the same power in 1/N. For the same reason, $u_{8}$ have to scale as $1/N^{3}$ and $u_{2p}$ as $1/N^{p-1}$. Finally, the division by 1/(2p)! ensures that the symmetry factors of the Feynman diagrams match exactly with the dimension of its own discrete symmetry group.

From (Equation 22), we easily deduce that
(24)$$G_{k,\mu \mu^{\prime }}=\frac{1}{p_{\mu }^{2}+u_{2}+r_{k}\left(p_{\mu }^{2}\right)}\;\delta_{p_{\mu },-p_{\mu^{\prime }}}\;.$$ To compute the flow equation, we have to make a choice for the regulator. From the expected form of the propagator, it is suitable to chose the Litim regulator—which is optimized in the sense of [67,68]:(25)$$r_{k}\left(p_{\mu }^{2}\right)=\left(k^{2}-p_{\mu }^{2}\right)\theta \left(k^{2}-p_{\mu }^{2}\right)\;,$$
where θ(x) denotes the standard Heaviside function. The flow equation for $u_{2}$ follows:(26)$${\dot{u}}_{2}=-\frac{1}{2N}\frac{2k^{2}}{{\left(k^{2}+u_{2}\right)}^{2}}\sum_{\mu }\theta \left(k^{2}-p_{\mu }^{2}\right)\Gamma_{k,\mu \mu \mu_{1}\mu_{1}}^{\left(4\right)}{|}_{p_{\mu_{1}}=0}\;.$$ In the large *N* limit, it is suitable to convert the sum as an integration, following [23]. For power law distributions $\rho \left(p^{2}\right)={\left(p^{2}\right)}^{\alpha }$, the resulting equations are exactly the same as for standard field theory in dimension *d*, for which $\rho \left(p^{2}\right)={\left(p^{2}\right)}^{d/2-1}$. The RG proceed usually in two steps. As a first step we integrate degrees of freedom into some range of momenta p∈[sΛ,Λ] (s<1), providing a change of cut-off Λ→sΛ. The second step is a global dilatation p→p/s, ensuring that the original UV cut-off Λ is restored (see Figure 5).

The shape of a power-law distribution is globally invariant to such transformations. The consequence is the existence of a global scale-dependent rescaling $u_{p}\to k^{d_{p}}{\bar{u}}_{p}$ for all couplings, such that flow equations become autonomous. In that equation, $d_{p}$ is the scaling (or canonical) dimension for $u_{p}$. The distribution that we consider in this paper, like MP law, do not enjoy this shape invariance property, and the flow equation never look as an autonomous system. The best compromise that we can do is a local definition of the canonical dimension, as in [23] with the example of quartic coupling. Here, we reproduce some parts of this analysis, providing a deeper investigation of the local scaling dimensions for higher couplings.

#### 3.1.2. Flow Equations, Scaling and Dimension

Because of the asymptotic nature for $u_{2}$, it is suitable to assume that it must scale as $k^{2}$, and following [23], we define the dimensionless mass as ${\bar{u}}_{2}=k^{-2}u_{2}$. Thus, without assumptions on the distribution ρ we get:(27)$${\dot{\bar{u}}}_{2}=-2{\bar{u}}_{2}-\frac{2u_{4}}{{\left(1+{\bar{u}}_{2}\right)}^{2}}\frac{1}{k^{2}}\int_{0}^{k}\rho \left(p^{2}\right)pdp\;,$$
with the notation $\dot{X}=kdX/dk$. For a power law distribution, $L:=\int_{0}^{k}\rho \left(p^{2}\right)pdp$ equals $L=k^{2\alpha +2}/\left(2\alpha +2\right)$; therefore
(28)dln(L)=(2α+2)dln(k). The variation of the loop integral is proportional to the variation of the time flow t=ln(k). This is why the parameter *t* is as well relevant for ordinary QFT. For ρ being not a power law however, it is suitable to use the time τ defined as dτ:=dL. In this parametrization we get straightforwardly:(29)$$\frac{d{\bar{u}}_{2}}{d\tau }=-2\frac{dt}{d\tau }{\bar{u}}_{2}-\frac{2u_{4}}{{\left(1+{\bar{u}}_{2}\right)}^{2}}\frac{\rho (k^{2})}{k^{2}}{\left(\frac{dt}{d\tau }\right)}^{2}\;,$$
and we define the τ-dimension for $u_{2}$
(30)$$\dim_{\tau }\left(u_{2}\right)=2\frac{dt}{d\tau }\;.$$ The τ-dimension for $u_{4}$ can be defined in the same way,
(31)$$u_{4}\frac{\rho (k^{2})}{k^{2}}{\left(\frac{dt}{d\tau }\right)}^{2}=:{\bar{u}}_{4}\;,$$
ensuring that the non autonomous character of the flow is entirely contained in the linear term of the flow equations. We obtain finally:(32)$$\frac{d{\bar{u}}_{2}}{d\tau }=-2\frac{dt}{d\tau }{\bar{u}}_{2}-\frac{2{\bar{u}}_{4}}{{\left(1+{\bar{u}}_{2}\right)}^{2}}\;.$$ For the coupling $u_{4}$, taking the fourth derivative of the flow Equation (Equation 18) and excluding the odd functions which vanish, we get:(33)$$\frac{du_{4}}{d\tau }=-\frac{2u_{6}}{{\left(1+{\bar{u}}_{2}\right)}^{2}}\rho \left(k^{2}\right){\left(\frac{dt}{d\tau }\right)}^{2}+\frac{12u_{4}^{2}}{{\left(1+{\bar{u}}_{2}\right)}^{3}}\;\frac{\rho (k^{2})}{k^{2}}{\left(\frac{dt}{d\tau }\right)}^{2}\;.$$ Thus, rescaling $u_{6}$ in such a way that only the linear term in ${\bar{u}}_{4}$ is scale-dependent enforces the definition:(34)$$u_{6}\;k^{2}{\left(\frac{\rho (k^{2})}{k^{2}}{\left(\frac{dt}{d\tau }\right)}^{2}\right)}^{2}=:{\bar{u}}_{6}\;.$$ Therefore:(35)$$\frac{d{\bar{u}}_{4}}{d\tau }=-\dim_{\tau }\left(u_{4}\right){\bar{u}}_{4}-\frac{2{\bar{u}}_{6}}{{\left(1+{\bar{u}}_{2}\right)}^{2}}+\frac{12{\bar{u}}_{4}^{2}}{{\left(1+{\bar{u}}_{2}\right)}^{3}}\;,$$
where:(36)$$\dim_{\tau }\left(u_{4}\right):=-2\left(\frac{t^{\prime \prime }}{t^{\prime }}+t^{\prime }\left(\frac{1}{2}\frac{d\ln\rho }{dt}-1\right)\right)\;,$$
denoting as $X^{\prime }$ for dX/dτ. Finally, we get for $u_{6}$:(37)$${\bar{u}}_{6}^{\prime }=-\dim_{\tau }\left(u_{6}\right){\bar{u}}_{6}+60\;\frac{{\bar{u}}_{4}{\bar{u}}_{6}}{{\left(1+{\bar{u}}_{2}\right)}^{3}}-108\;\frac{{\bar{u}}_{6}^{3}}{{\left(1+{\bar{u}}_{2}\right)}^{4}}\;;$$
where:(38)$$-\dim_{\tau }\left(u_{6}\right):=2\frac{dt}{d\tau }+4\left(\frac{t^{\prime \prime }}{t^{\prime }}+t^{\prime }\left(\frac{1}{2}\frac{d\ln\rho }{dt}-1\right)\right)\;.$$ In the same way, we get for $u_{2p}$:(39)$$-\dim_{\tau }\left(u_{2p}\right)=2\left(p-2\right)\frac{dt}{d\tau }-\left(p-1\right)\dim_{\tau }\left(u_{4}\right)\;.$$

#### 3.1.3. Analytical Noisy Data, MP Distribution

For our numerical investigations we need to keep control of the size of the signal and numerical approximations. To this end, we consider deformations around a model of noise. We focus on the MP law, which has the double advantage to be a familiar model of noise and to be analytic. For *X*
N×P matrix with i.i.d. entries and variance $\sigma^{2}<\infty$, the MP distribution μ(x) gives the spectrum of the correlation matrix $Z:=\frac{X^{T}X}{P}$ for both N,P→∞ but P/N=:K remains finite [43]. Explicitly:(40)$$\mu \left(x\right)=\frac{1}{2\pi \sigma^{2}}\frac{\sqrt{\left(a_{+}-x\right)\left(x-a_{-}\right)}}{Kx}\;,$$
where $a_{\pm }=\sigma^{2}{\left(1\pm \sqrt{K}\right)}^{2}$. The distribution ρ for eigenvalues of the inverse matrix can be easily deduced from (Equation 40). Figure 6 provides a picture of the numerical flow for a quartic truncation. Interestingly, the behavior of the RG flow looks very close to the familiar flow for $\phi^{4}$ theory in dimension d<4. In particular, we show the existence of two regions, one in which the flow goes toward positive mass and the second one toward the negative mass. Usually, this splitting is governed by a fixed point, the Wilson–Fisher fixed point. Even though we have no true fixed point, in this case, we show that an analogous effect appears, the role of the fixed point being played by an extended attractive region that we call pseudo-fixed point.

In Figure 7 we plotted the canonical dimensions of the couplings up to p=5, for K=1 and σ=0.5,1 and 2, respectively. This is the property announced in Section 2.1. In the deep UV sector, i.e., in the domain of very small eigenvalues, the canonical dimension is positive for an arbitrarily large number of interactions. In the RG language, this means that an arbitrarily large number of operators are relevant toward the IR scales, and the description of the flow becomes very difficult, requiring to consider very large truncations in a very small range of scales. In contrast, up to a scale, $\Lambda_{0}\left(\sigma \right)$, defined such that:(41)$${\left[\frac{dt}{d\tau }-\frac{3}{4}\dim_{\tau }\left(u_{4}\right)\right]}_{t=\ln(\Lambda_{0})}=0\;,$$
only the local couplings $u_{4}$ and $u_{6}$ are relevant. Numerically, this point is reached in the vicinity of the eigenvalue $\lambda ∼\lambda_{0}/3$, $\lambda_{0}$ denoting the largest eigenvalue of the analytic spectrum. We, thus, have essentially revealed the existence of two regions: the *deep noisy region* (DNR), for $p^{2}>\Lambda_{0}$, where the number of relevant operators and their respective canonical dimensions increases arbitrarily, and the *learnable region* (LR) for $p^{2}<\Lambda_{0}$, where only two couplings are relevant and standard field theoretical methods are expected to work. In this paper we only focus on this region; using RG to track the presence of a signal. Figure 8 shows numerical evolution of couplings $u_{2}$, $u_{4}$, and $u_{6}$, starting the RG flow from $k=\Lambda_{0}$. Note, finally, that the behavior of the canonical dimension can be expected from the small *p* behavior of the MP law. Indeed for small *p*, $\rho ∼{\left(p^{2}\right)}^{\alpha }$ with α=1/2. Following the dimensional analysis in [23], the corresponding canonical dimension for the local couplings $u_{2p}$ must be $\dim_{t}\left(u_{2p}\right)=2\left(1-\left(p-1\right)\alpha \right)$, and, thus, interactions are irrelevant for p>3. The asymptotic behavior of the distribution provides, therefore, a first indication of the relevant interactions in the asymptotic region, and we call *critical dimension* the corresponding value for α.

#### 3.1.4. A First Look on Numerical Investigations

After the previous analytic observations, we provide in this section a first inspection on numerical aspects for non-analytic signals, as illustrated in Figure 9.

To keep control on the strength of the signal, we focus on two kind of eigenvalue distributions for our numerical investigations. The first one is a model of noise, that we call *NMP* (numerical Marchenko–Pastur). It is obtained by constructing the covariance matrix for a N×P matrix with i.i.d. random entries and variance equal to 1. The distribution of the eigenvalues of such matrix converges, for large *P* and *N*, to the MP’s law, and we set P=1500 and N=2000 for all our simulations that we discuss in this section. The second distribution that we call *DNMP* (disturbed numerical Marchenko–Pastur) is obtained from the first one by adding the spikes associated to a matrix of rank R=65 (defining the size of the signal). The variance being fixed to 1, the canonical dimensions for the purely noisy data are given by Figure 7. We focus on the learnable region (LR), for eigenvalues between 2.5 and 3.4, where only the $\phi^{4}$ and $\phi^{6}$ interactions are relevant.

In [23], the authors pointed out that the canonical dimension of relevant operators decreases with the presence of a signal in the LR, and in particular for strong enough signal, [g] becomes negative. Therefore, we expect the existence of a sufficiently small neighbourhood near the Gaussian fixed point where the field theory goes toward an asymptotic Gaussian behavior, arbitrarily close to the mass axis. This heuristic behavior based on dimensional considerations illustrates how the presence of the signal can affect the equivalence class of asymptotic states. Here, we are aiming to go beyond these dimensional considerations, and investigate non-perturbative effects with regard to the field expectation value.

Figure 10 shows the RG flow for the NMP law disturbed by a signal materialized with discrete spikes. Comparing with the purely MP law (Figure 6), we show that the pseudo-fixed point moved toward the Gaussian fixed point. This illustrate how RG may be used to track the presence of a signal. Indeed, the (pseudo-)fixed point controls the critical behavior. If its position changes, one expects that IR physics may be affected for some initial conditions. Among these IR properties, we focus in this paper on the field vacuum expectation value. In the truncation that we considered, and neglecting the momentum dependence of the classical field, the effective potential writes as a sixtic polynomial:(42)$$U\left(m,\left\{u_{2n}\right\}\right)=\frac{1}{2}u_{2}m^{2}+\frac{u_{4}}{4!}\;m^{4}+\frac{u_{6}}{6!}\;m^{6}\;,$$
up to the rescaling M=:Nm. The classical configuration is such that ∂U/∂m=0, and it depends on the values and on the signs of the different couplings. We focus on the sixtic truncations, and in the region $u_{6}>0$, ensuring integrability. Under these conditions, we investigate, in the vicinity of the Gaussian fixed point, the set of initial conditions ending in the symmetric phase, i.e., such that the values of the couplings ensure m=0. The set of these points takes the form of a compact region $R_{0}$ around the Gaussian point.

Figure 11 shows this compact region $R_{0}$ for a purely noisy data (on the top) and for a small perturbation of the previous one with a multi-spike signal (on the bottom). Figure 12 illustrates what happens in regard to the shape of the effective potential. On the left, we show the evolution of the effective potential in the purple region for a purely noisy data. On the right, we present the evolution of the potential for the same initial conditions, when the noise is disturbed with some spikes.

This observation highlights a strong equivalence between the presence of a signal and the lack of symmetry restoration for some RG trajectories. However, a moment of reflection shows that region $R_{0}$ is too large, and that physically relevant states have to be researched as a compact subset $\varepsilon_{0}$ of this region constructed as the intersection of $R_{0}$ with the constraint imposed to the probability distribution in the IR. The relevant one in the LPA is that the mass $u_{2}$ (interpreted as the largest eigenvalue) remains finite, implying that ${\bar{u}}_{2}$ scales as $k^{-2}$ for small *k*. Since the separation between two eigenvalues is of order 1/N, one expects that the smallest value for $k^{2}$ is ∼1/*N*. Figure 13 shows that trajectories satisfying this requirement exist in $R_{0}$, and $\varepsilon_{0}\neq \emptyset$.

The existence of a non-vanishing set of physically relevant states grants the possibility to propose the following scenario. We showed that the size of the region $R_{0}$ decreases due to the presence of a signal in the spectrum. However, as long as this collapse does not affect the subset $\varepsilon_{0}$, the presence of the signal has no relevant consequences with respect to the physical states, at least concerning the expectation value of the field. Our observations suggest that this is happening only for a signal which is large enough, thus providing evidence in favor of the existence of an intrinsic detection threshold working with the expectation value. We do not address the issue of the precise determination of the shape of this subset $\varepsilon_{0}$, which is the purpose of the companion paper [24].

### 3.2. Venturing into the Non-Symmetric Phase

#### 3.2.1. LPA and LPA${}^{\prime }$

In this section, we consider the LPA and its improved version LPA${}^{\prime }$. This way, our assumptions about $\Gamma_{k,\mathrm{kin}}$ (Equation (Equation 19)) hold, but we include the mass contribution into the local potential $U_{k}\left[M\right]$. Moreover, we neglect the momentum dependence of the classical field M(p), dominated by the zero-momentum (large scale) value:(43)$$M\left(p\right)∼M\delta_{p0}\;.$$ This approximation usually holds in the IR region, which is exactly what we consider. Moreover, it is not hard to show that such an expansion around M=0 reproduces exactly the same equations as the truncation (Equation 21) for local operators (i.e., neglecting the momentum dependence of the effective vertices $\Gamma_{k}^{\left(2p\right)}$). This approximation works well at large scale, where a symmetry breaking scenario is expected, requiring an expansion around a non-vanishing vacuum M≠0. For this reason, we consider the following parametrization:(44)$$U_{k}\left[\chi \right]=\frac{u_{4}\left(k\right)}{2!}{\left(\chi -\kappa (k)\right)}^{2}+\frac{u_{6}\left(k\right)}{3!}{\left(\chi -\kappa (k)\right)}^{3}+\cdots \;,$$
where $\chi :=M^{2}/2$, and κ(k) is the running vacuum. The global normalization is such that, for $M_{0}\left(p\right)=M\delta_{p0}$, $\Gamma_{k}\left[M=M_{0}\right]=NU_{k}\left[\chi \right]$. The 2-point vertex $\Gamma_{k}^{\left(2\right)}$ moreover is defined as:(45)$$\Gamma_{k,\mu \mu^{\prime }}^{\left(2\right)}=\left(Z\left(k\right)p^{2}+\frac{\partial^{2}U_{k}}{\partial M^{2}}\right)\delta_{p_{\mu },-p_{\mu^{\prime }}}\;,$$
and, thus, replaces the formula (Equation 22), the role of the mass being played by the second derivative of the potential. The flow equation for $U_{k}$ can be deduced from (Equation 18), setting $M=M_{0}$ on both sides. Assuming, once again, that *N* is large and using the continuum setting, we get:(46)$${\dot{U}}_{k}\left[M\right]=\frac{1}{2}\;\int pdp\;k\partial_{k}\left(r_{k}\left(p^{2}\right)\right)\rho \left(p^{2}\right)\left(\frac{1}{\Gamma_{k}^{\left(2\right)}+r_{k}}\right)\left(p,-p\right)\;.$$ Note that in the definition (Equation 45) we introduced the anomalous dimension Z(k), which has a non-vanishing flow equation for κ≠0. To take into account the non vanishing flow for *Z*, it is suitable to slightly modify the Litim regulator as:(47)$$r_{k}\left(p^{2}\right)=Z\left(k\right)\left(k^{2}-p^{2}\right)\theta \left(k^{2}-p^{2}\right)\;.$$ This modification simplifies the computation of the integrals [67,68]. In the computation of the flow equations, however it is suitable to rescale the dimensionless couplings ${\bar{u}}_{2p}\to Z^{-p}{\bar{u}}_{2p}$, such that the coefficient in front of $p^{2}$ in the kinetic action remains equal to 1. This additional rescaling adds a term nη(k) in the flow equation, where η, the *anomalous dimension* is defined as:(48)$$\eta \left(k\right)=\frac{\dot{Z}\left(k\right)}{Z(k)}\;.$$ Despite the fact that it simplifies the computation, the factor *Z* in front of the regulator (Equation 47) must not affect the boundary conditions $\Gamma_{k=\infty }\to S$ and $\Gamma_{k=0}\to \Gamma$. In particular, the first one requires that $r_{k\gg 1}∼k^{r}$, for positive *r*. This is obviously the case for Z=1, $r_{k\gg 1}∼k^{2}$. However, it is possible for *Z* to break this condition. This may be the case, for instance, if the flow reaches a fixed point *p*. At this point, the anomalous dimension takes a value $\eta_{p}$, thus $Z\left(k\right)=k^{\eta_{p}}$ and $r_{k\gg 1}∼k^{2+\eta_{p}}$. The requirement r>0 then imposes $\eta_{p}>-2$. Obviously, this is a limitation of the regulator, not of the method. Moreover, the non-autonomous nature of the RG equation prevents the existence of exact fixed points, so that the criteria should be more finely defined. Generally, one expects that the LPA approximation makes sense only in regimes where η is not so large, and becomes spurious in regime where |η|≳1 [71].

RG equation for η=0

As a first approximation, standard LPA sets Z(k)=1, or equivalently η=0. From (Equation 46), we arrive to the expression:(49)$${\dot{U}}_{k}\left[\chi \right]=\left(2\int_{0}^{k}\rho \left(p^{2}\right)pdp\right)\;\frac{k^{2}}{k^{2}+\partial_{\chi }U_{k}\left(\chi \right)+2\chi \partial_{\chi }^{2}U_{k}\left(\chi \right)}\;.$$ Introducing the flow parameter τ defined in Section 3.1, we get:(50)$$U_{k}^{\prime }\left[\chi \right]=k^{2}\rho \left(k^{2}\right){\left(\frac{dt}{d\tau }\right)}^{2}\;\frac{k^{2}}{k^{2}+\partial_{\chi }U_{k}\left(\chi \right)+2\chi \partial_{\chi }^{2}U_{k}\left(\chi \right)}\;,$$ First, we define the scaling of the effective potential as:(51)$$\partial_{\chi }U_{k}\left(\chi \right)k^{-2}=\partial_{\bar{\chi }}{\bar{U}}_{k}\left(\bar{\chi }\right)\;,\;\chi \partial_{\chi }^{2}U_{k}\left(\chi \right)k^{-2}=\bar{\chi }\partial_{\bar{\chi }}^{2}{\bar{U}}_{k}\left(\bar{\chi }\right)\;,$$
therefore:(52)$$U_{k}^{\prime }\left[\chi \right]={\left(\frac{dt}{d\tau }\right)}^{2}\;\frac{k^{2}\rho \left(k^{2}\right)}{1+\partial_{\bar{\chi }}{\bar{U}}_{k}\left(\bar{\chi }\right)+2\bar{\chi }\partial_{\bar{\chi }}^{2}{\bar{U}}_{k}\left(\bar{\chi }\right)}$$ The Equation (Equation 51) fixes the relative scaling of $U_{k}$ and χ. The previous relation moreover fixes the absolute scaling (the word “absolute” simply means that all the flow equations remain invariant) under a global reparametrization. This property, moreover, can be read directly in the partition function, and it reflects the invariance of the path integral measure of $U_{k}$:(53)$$U_{k}\left[\chi \right]:={\bar{U}}_{k}\left[\bar{\chi }\right]k^{2}\rho \left(k^{2}\right){\left(\frac{dt}{d\tau }\right)}^{2}\;.$$ In order to find the appropriate rescaling for χ, we introduce a scale dependent factor *A*, and define $\bar{\chi }$ as $\chi =A\bar{\chi }$. From global coherence, $\bar{\chi }$ has to be such that:(54)$$U_{k}\left[\chi \right]:={\bar{U}}_{k}\left[A^{-1}\chi \right]k^{2}\rho \left(k^{2}\right){\left(\frac{dt}{d\tau }\right)}^{2}\;.$$ Therefore, expanding in power of χ, we find that the linear term becomes:(55)$$\partial_{\chi }U_{k}\left(\chi =0\right)\chi =\partial_{\bar{\chi }}{\bar{U}}_{k}\left[\bar{\chi }=0\right]\bar{\chi }k^{2}\rho \left(k^{2}\right){\left(\frac{dt}{d\tau }\right)}^{2}\;,$$
or, from (Equation 51):(56)$$\partial_{\chi }U_{k}\left(\chi =0\right)\chi =\partial_{\chi }U_{k}\left(\chi =0\right)\chi A^{-1}\rho \left(k^{2}\right){\left(\frac{dt}{d\tau }\right)}^{2}\;.$$ Then, assuming $\partial_{\chi }U_{k}\left(\chi =0\right)\chi \neq 0$, we get:(57)$$A=\rho \left(k^{2}\right){\left(\frac{dt}{d\tau }\right)}^{2}\;,$$
and:(58)$$\chi =\rho \left(k^{2}\right){\left(\frac{dt}{d\tau }\right)}^{2}\bar{\chi }\;.$$ This equation, obviously fixes the dimension of κ which must be the same as χ. The flow equations for the different couplings must be derived from definition: (59)$$\frac{\partial U_{k}}{\partial \chi }{|}_{\chi =\kappa }=0\;,$$
(60)$$\frac{\partial^{2}U_{k}}{\partial \chi^{2}}{|}_{\chi =\kappa }=u_{4}\left(k\right)\;,$$
(61)$$\frac{\partial^{3}U_{k}}{\partial \chi^{3}}{|}_{\chi =\kappa }=u_{6}\left(k\right)\;.$$ The first equation is nothing but the mathematical translation of the requirement that the expansion is made around a local minimum. The two other equations are consequence of the parametrization of $U_{k}$. In order to derive the flow equations for dimensionless couplings, it is suitable to work with a flow equation at fixed $\bar{\chi }$ rather than fixed χ:(62)$$\begin{array}{cc} U_{k}^{\prime }\left[\chi \right]=\rho \left(k^{2}\right){\left(\frac{dt}{d\tau }\right)}^{2} & \left[{\bar{U}}_{k}^{\prime }\left[\bar{\chi }\right]+\dim_{\tau }\left(U_{k}\right){\bar{U}}_{k}\left[\bar{\chi }\right]-\dim_{\tau }\left(\chi \right)\bar{\chi }\frac{\partial }{\partial \bar{\chi }}{\bar{U}}_{k}\left[\bar{\chi }\right]\right]\;, \end{array}$$
where $\dim_{\tau }\left(U_{k}\right)$ and $\dim_{\tau }\left(\chi \right)$ denote, respectively, the canonical dimension of $U_{k}$ and χ, respectively. To compute them, we return on their definitions, explicitly:(63)$$\dim_{\tau }\left(U_{k}\right)=t^{\prime }\frac{d}{dt}\ln\left(k^{2}\rho \left(k^{2}\right){\left(\frac{dt}{d\tau }\right)}^{2}\right)\;,$$
and
(64)$$\dim_{\tau }\left(\chi \right)=t^{\prime }\frac{d}{dt}\ln\left(\rho \left(k^{2}\right){\left(\frac{dt}{d\tau }\right)}^{2}\right)\;.$$ The final expression for the effective potential RG equation then becomes:(65)$$\begin{array}{cc} {\bar{U}}_{k}^{\prime }\left[\bar{\chi }\right]= & -\dim_{\tau }\left(U_{k}\right){\bar{U}}_{k}\left[\bar{\chi }\right]+\dim_{\tau }\left(\chi \right)\bar{\chi }\frac{\partial }{\partial \bar{\chi }}{\bar{U}}_{k}\left[\bar{\chi }\right]+\;\frac{1}{1+\partial_{\bar{\chi }}{\bar{U}}_{k}\left(\bar{\chi }\right)+2\bar{\chi }\partial_{\bar{\chi }}^{2}{\bar{U}}_{k}\left(\bar{\chi }\right)}\;. \end{array}$$ The next steps are standard. From the definition (Equation 59) we must have $\partial_{\bar{\chi }}{\bar{U}}_{k}^{\prime }\left[\bar{\chi }=\bar{\kappa }\right]=-{\bar{u}}_{4}\;{\bar{\kappa }}^{\prime }$. Thus, taking the second derivative of (Equation 65), we get for ${\bar{\kappa }}^{\prime }$:(66)$${\bar{\kappa }}^{\prime }=-\dim_{\tau }\left(\chi \right)\bar{\kappa }+2\frac{3+2\bar{\kappa }\frac{{\bar{u}}_{6}}{{\bar{u}}_{4}}}{{\left(1+2\bar{\kappa }{\bar{u}}_{4}\right)}^{2}}$$ In the same way, taking second and third derivatives, and from the conditions (Equation 60) and (Equation 61), we get:(67)$$\begin{array}{cc} {{\bar{u}}_{4}}^{\prime }=-\dim_{\tau }\left(u_{4}\right){\bar{u}}_{4} & +\dim_{\tau }\left(\chi \right)\bar{\kappa }{\bar{u}}_{6}-\;\frac{10{\bar{u}}_{6}}{{\left(1+2\bar{\kappa }{\bar{u}}_{4}\right)}^{2}}+4\;\frac{{\left(3{\bar{u}}_{4}+2\bar{\kappa }{\bar{u}}_{6}\right)}^{2}}{{\left(1+2\bar{\kappa }{\bar{u}}_{4}\right)}^{3}}\;, \end{array}$$
and
(68)$$\begin{array}{cc} {{\bar{u}}_{6}}^{\prime }=-\dim\left(u_{6}\right){\bar{u}}_{6} & -12\;\frac{{\left(3{\bar{u}}_{4}+2\bar{\kappa }{\bar{u}}_{6}\right)}^{3}}{{\left(1+2\bar{\kappa }{\bar{u}}_{4}\right)}^{4}}+40{\bar{u}}_{6}\;\frac{3{\bar{u}}_{4}+2\bar{\kappa }{\bar{u}}_{6}}{{\left(1+2\bar{\kappa }{\bar{u}}_{4}\right)}^{3}}\;. \end{array}$$

The flow equation for η

We now assume that η(k)≠0. From definition, assuming that *Z* depends only on the value of the vacuum, we must have:(69)$$Z\left[M=\kappa \right]\equiv \frac{d}{dp^{2}}\Gamma_{k}^{\left(2\right)}\left(p,-p\right){|}_{M=\sqrt{2\kappa }}\;.$$ Therefore:(70)$$\eta \left(k\right):=\frac{1}{Z}k\frac{dZ}{dk}=\frac{1}{Z}\frac{d}{dp^{2}}{\dot{\Gamma }}_{k}^{\left(2\right)}\left(p,-p\right)\;.$$ The flow equation for $\Gamma_{k}^{\left(2\right)}$ can be deduced from (Equation 18), taking the second derivative with respect to the classical field. Due to the fact that, the effective vertex are momentum independent, in the LPA representation, the contributions involving $\Gamma_{k}^{\left(4\right)}$ have to be discarded from the flow equation for *Z*. Finally:(71)$$\dot{Z}:={\left(\Gamma_{k,000}^{\left(3\right)}\right)}^{2}\frac{d}{dp^{2}}\sum_{q}{\dot{r}}_{k}\left(q^{2}\right)G^{2}\left(q^{2}\right)G\left({\left(q+p\right)}^{2}\right){|}_{M=\sqrt{2\kappa },p=0}\;,$$
where, according to LPA, we evaluate the right hand side over uniform configurations. Therefore, $G\left(p,p^{\prime }\right)=:G\left(p\right)\delta \left(p+p^{\prime }\right)$ is the inverse of $\Gamma_{k}^{\left(2\right)}\left(p,p^{\prime }\right)+r_{k}\left(p^{2}\right)\delta \left(p+p^{\prime }\right)$, with $\Gamma_{k}^{\left(2\right)}$ given by equation (Equation 45). The expression of $\Gamma_{k,000}^{\left(3\right)}$ can be easily obtained; taking the third derivative of the effective potential for *M*:(72)$$\Gamma_{k,000}^{\left(3\right)}=3u_{4}\sqrt{2\kappa }+u_{6}{\left(2\kappa \right)}^{3/2}\;.$$ We arrive to the following expression for anomalous dimension (see Appendix A):(73)$$\eta \left(k\right)=2{\left(t^{\prime }\right)}^{-2}\frac{{\left(3\sqrt{2\bar{\kappa }}{\bar{u}}_{4}+{\left(2\bar{\kappa }\right)}^{3/2}{\bar{u}}_{6}\right)}^{2}}{{\left(1+2\bar{\kappa }{\bar{u}}_{4}\right)}^{4}}\;.$$ Note that, to derive this expression we have to take into account that the additional rescaling coming from *Z* accordingly to the requirement that the coefficient in front of $p^{2}$ in the kinetic action remains equals to 1. Requiring $\bar{\kappa }\to Z^{-1}\bar{\kappa }$ with respect to the strict LPA definition. Due to the factors *Z* in the definition of barred quantities, η(k) appears in the flow equations. The net result is a translation of canonical dimensions
(74)$$\dim_{\tau }\left(u_{2n}\right)\to \dim_{\tau }\left(u_{2n}\right)-n\frac{dt}{d\tau }\eta \left(k\right)$$
in the equations obtained within strict LPA.

#### 3.2.2. Numerical Investigations

The main goal in this section is to show that the general behavior that we observed for the DE in the symmetric phase holds using the LPA formalism, expanding around a non-zero vacuum. Figure 14 shows the existence of some RG trajectories for which the symmetry is restored within the range where the eigenvalues are between 2.5 and 3.4 (corresponding to the range where only the $\phi_{4}$ and $\phi_{6}$ interactions are relevant for the MP distribution with σ=1 and K=0.75). This is manifested by the fact that κ decreases to zero. We also show in the same figure that there are other RG trajectories which do not allow a restoration of the symmetry. Once again, we can identify a set of initial conditions in the vicinity of the Gaussian fixed point where symmetry is always restored in the deep IR. Furthermore, we show that there are initial coupling conditions that are of great interest for signal detection. In fact, for these initial conditions, we have a restoration of the symmetry when we consider data without signal and, conversely, we do not have such restoration when we add the signal in the data. This is illustrated in Figure 15 in the form of potentials for a specific initial coupling condition. Finally, we emphasize that there is no significant change in this general behavior when we apply the LPA’ representation instead of the LPA one, i.e., when we take into account the non-zero anomalous dimension (η) in the formalism. Indeed, we show in Figure 16, that this anomalous dimension remains very small for the range of eigenvalues that we consider. This moreover is expected to be a good indication of the convergence of the derivative expansion [71], which improves the reliability of our conclusions.

## 4. Concluding Remarks and Open Issues

Let us summarize our investigations in this paper:In order to keep control on the size of the signal and numerical approximations, we constructed datasets as perturbations around the MP law. We showed that the field theory approximation works well up to some scale $\Lambda_{0}$. From this scale, the relevant sector, spanned by relevant couplings, diverges (its dimension becomes arbitrarily large, and couplings have arbitrary large dimension), and we expect that standard approximation fails up to this scale.Above the scale $\Lambda_{0}$, and focusing on the local interactions, the relevant sector has dimension 2, spanned by $\phi^{4}$ and $\phi^{6}$ interactions, in agreement with a naive power counting based on the critical dimension α=1/2 of the MP law;For MP distribution, we showed the existence of a compact region $R_{0}$ in the vicinity of the Gaussian fixed point, whose RG trajectories end in the symmetric region, and thus are compatible with symmetry restoration scenario;Disturbing the MP spectrum with a strong enough signal reduces the size of this compact region, continuously deforming the effective potential from a symmetric toward a broken shape. In that picture, the role played by the signal strength is reminiscent of the role played by the inverse temperature β:=1/T in the physics of phase transition;Finally, considering intersection $\varepsilon_{0}$ between $R_{0}$ and the physical conditions imposed to the IR 2-point function, we provided evidence in favor of the existence of an intrinsic detection threshold in the LPA approximation. This region is fully investigated in the companion paper [24].

These conclusions have to be completed by some important remarks concerning the different approximations that we did, and by perspectives for forthcoming works.

The first one is about the approximation procedure used to solve the RG Equation (Equation 18). Indeed, despite the limitations of the field theory approximation, the standard recipes to solve RG equations present limitations. In particular, the LPA neglects the momentum dependence of the coupling (i.e., deviations from the strict local approximation). We have no doubts that such an approximation makes sense in the deep IR regime, at the tail of the spectrum, where momenta are weak, and non-local interactions appear less relevant than local ones. However, as we explore the small eigenvalue scales, the effect of derivative couplings can no longer be neglected. As long as these terms can be treated as corrections, it is expected that our conclusions will not change significantly. However, these corrections could play a role in the estimation of the detection criterion. Note that, in regimes where momenta take large values and DE breaks down, other approximation schemes exist, enabling to keep the full dependence of the effective vertices. The most popular being the so-called Blaizot–Mendez–Wschebor (BMW) method [21,22,32,33,34,35,36,37,38,39,40,41,42,43,44,45,46,47,48,49,50,51], which, combined with exact relations such as Ward identities, gives the possibility to provide exact (i.e., scheme independent) results [61]. Another current source of disagreement concerns the choice of the regulator. However, our conclusions are based on the behavior of the effective potential rather than on a specific value of a physically relevant quantity as a critical exponent. We thus expect them to be solid with respect to this issue [62,63,64,72,72].

The other source of approximation is the theoretical embedding. We showed that such an embedding offers a satisfactory description only for a small eigenvalue region. As we pointed out, such a limitation is not a novelty in physics, and it may be the sign that a more fundamental description has to replace the field theory approximation. Equation (Equation 8), involving discrete spins, provides an example of such a description. Note that, conversely, our field theory can be viewed as an effective description of such a binary model described with a constrained maximum entropy distribution. Finally, we focused on equilibrium, i.e., maximum entropy states, although we have adopted the standard field theory view of the RG. With this respect, let us mention the interesting possibility to view the exact RG equations as a form of entropy dynamics [40].

Our results focused on a specific model of noise, closer to the analytical MP law. Obviously, this is far from exhausting the large diversity of models. One might expect our conclusions to be much more general, and that they could be a universal property of all statistical noise models. This conjecture however has to be supported by deeper investigations, and we plan to address this issue for other models of noise. A first step in this direction has also been done recently for data materialized by random tensors rather than matrices [25], as considered in the topic of tensorial PCA [73].

## Figures and Tables

**Figure 1 entropy-23-01132-f001:**
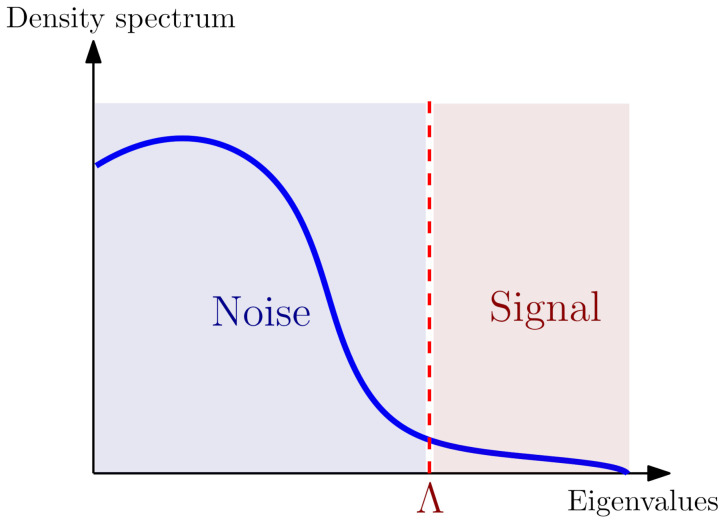
Qualitative picture of the signal detection issue in a nearly continuous spectrum.

**Figure 2 entropy-23-01132-f002:**
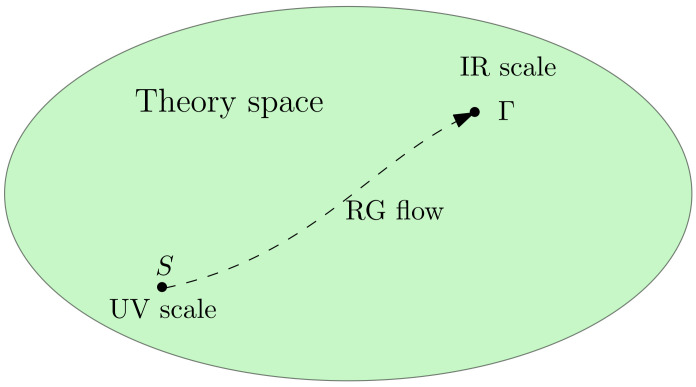
Qualitative illustration of the RG flow. The UV scale is described by the classical action S, while the IR scale is described by an effective object Γ, where microscopic effects are hidden in the different parameters involved in its definition.

**Figure 3 entropy-23-01132-f003:**
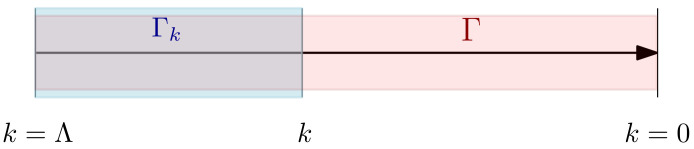
Qualitative illustration of the meaning of the effective averaged action $\Gamma_{k}$, as the effective action of the UV degrees of freedom which have been integrated-out.

**Figure 4 entropy-23-01132-f004:**
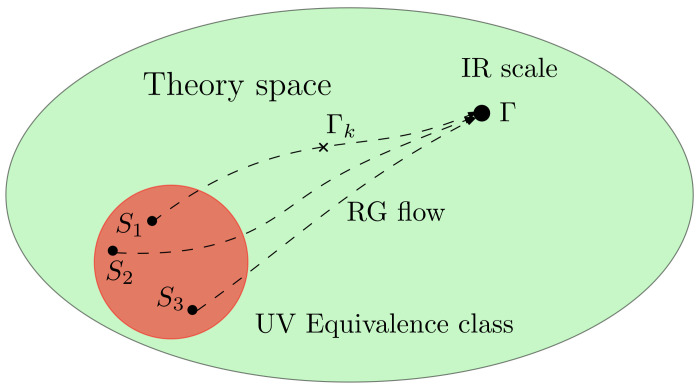
Qualitative illustration of the RG flow behavior. Some different UV initial conditions lead to the same (universal) IR physics, up to negligible differences with regard to the experimental precision.

**Figure 5 entropy-23-01132-f005:**
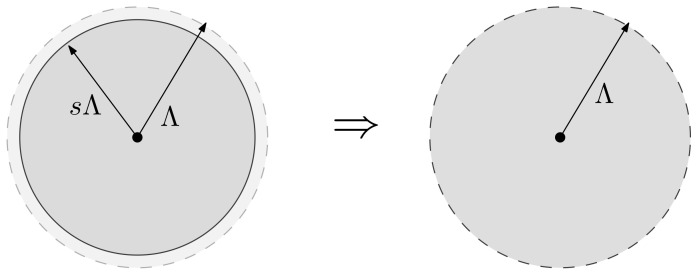
A step of the RG flow. On the left, integration of momenta between sΛ and Λ. On the right, dilatation of the remaining momenta with a factor 1/s.

**Figure 6 entropy-23-01132-f006:**
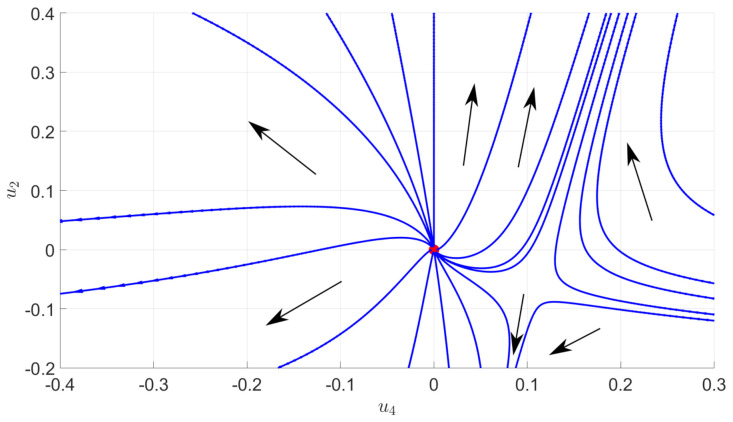
Numerical flow associated to the MP law (data without signal) and for the quartic truncation. The main directions of the flow are highlighted by the black arrows (which are oriented from UV to IR). We observe the existence of a region reminiscent of the standard Wilson–Fisher fixed point.

**Figure 7 entropy-23-01132-f007:**
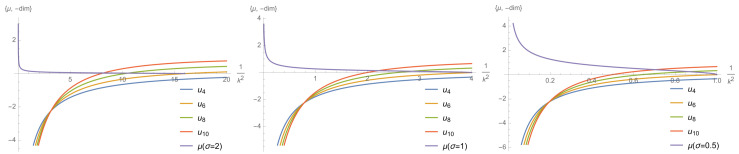
The canonical dimension for MP distribution with K=1 and σ=0.5 (on the right), σ=1 (in the middle) and σ=2 (on the left). The purple curve corresponds to the MP distribution.

**Figure 8 entropy-23-01132-f008:**
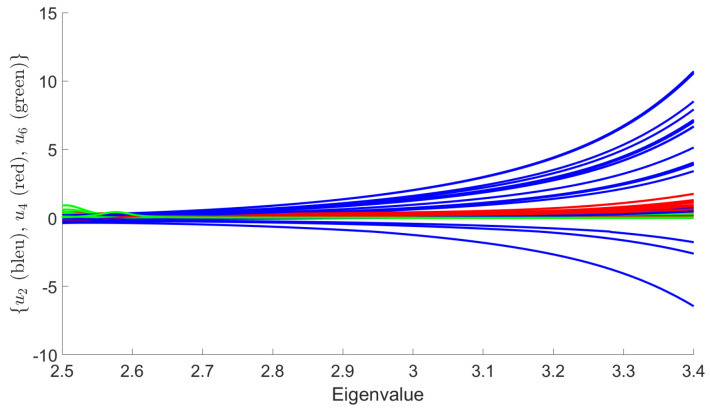
RG trajectories starting from $k=\Lambda_{0}$ for $u_{2}$ (blue curves), $u_{4}$ (red curves), and $u_{6}$ (green curves).

**Figure 9 entropy-23-01132-f009:**
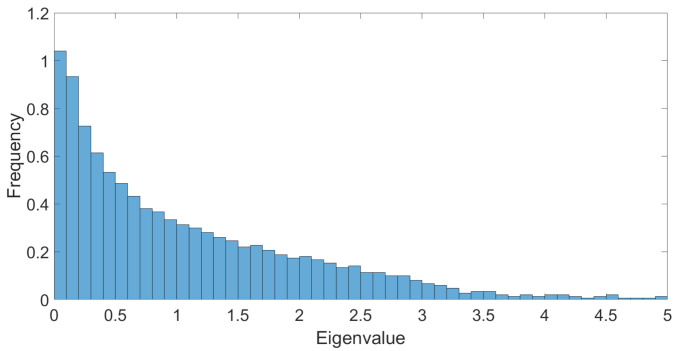
A typical DNMP distribution for P=1500 and N=2000.

**Figure 10 entropy-23-01132-f010:**
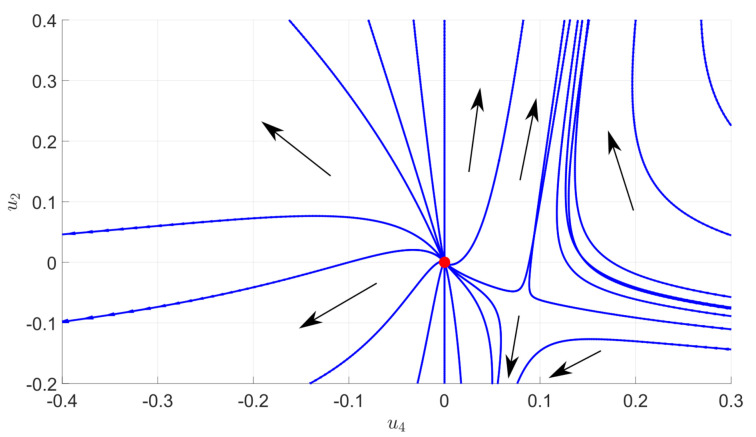
Numerical flow associated to a DNMP distribution in the learnable region.

**Figure 11 entropy-23-01132-f011:**
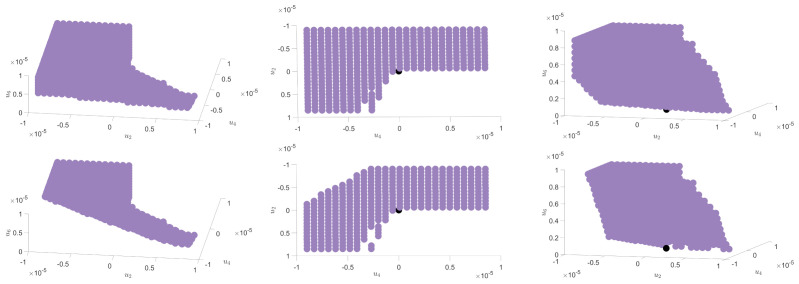
Three points of view of the compact region $R_{0}$ (illustrated with purple dots) in the vicinity of the Gaussian fixed point (illustrated with a black dot). In this region RG trajectories end in the symmetric phase, and thus are compatible with a symmetry restoration scenario for initial conditions corresponding to an explicit symmetry breaking. The top plots are associated to the case of pure noise and the bottom plots are, respectively, associated to the case with signal.

**Figure 12 entropy-23-01132-f012:**
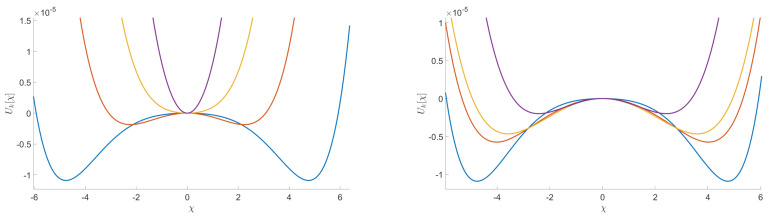
Illustration of the evolution of the potential associated to an example of initial conditions of the coupling $u_{2}$, $u_{4}$, and $u_{6}$ where the RG trajectories end in the symmetric phase in the case of pure noise (on the left) and stay in the non-symmetric phase when we add a signal (on the right).

**Figure 13 entropy-23-01132-f013:**
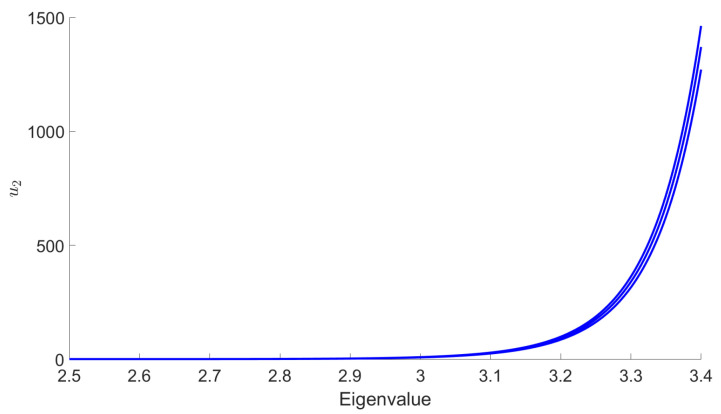
Illustration of the evolution of the $u_{2}$ for eigenvalues between 2.5 and 3.4 in the case of pure noise (NMP distribution). We can see that the values of ${\bar{u}}_{2}$ for these examples are of the same magnitude as N=2000. This highlights the existence of some RG trajectories associated to physically relevant states in the deep infrared.

**Figure 14 entropy-23-01132-f014:**
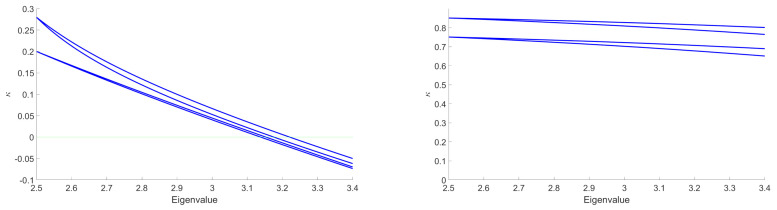
Illustration of the evolution of κ, obtained with the LPA representation, for eigenvalues between 2.5 and 3.4 in the case of data without signal. For some RG trajectories (on the left), κ decreases to zero, which correspond to a restoration of the symmetry. For other RG trajectories (on the right), κ stays almost constant in the range of eigenvalues that we consider, and does not lead to a restoration of the symmetry.

**Figure 15 entropy-23-01132-f015:**
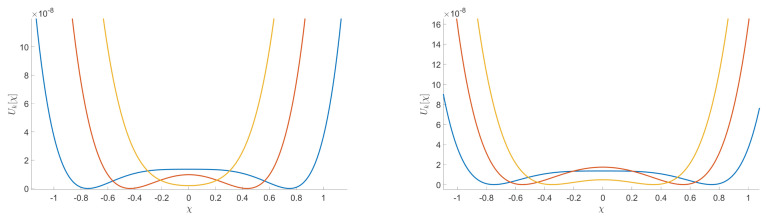
Illustration of the evolution of the potential associated to an example of initial conditions of the coupling $u_{2}$, $u_{4}$, and $u_{6}$. We see that the RG trajectories, obtained with the LPA representation, end in the symmetric phase in the case of pure noise (on the left) and stay in the non symmetric phase when we add a signal (on the right).

**Figure 16 entropy-23-01132-f016:**
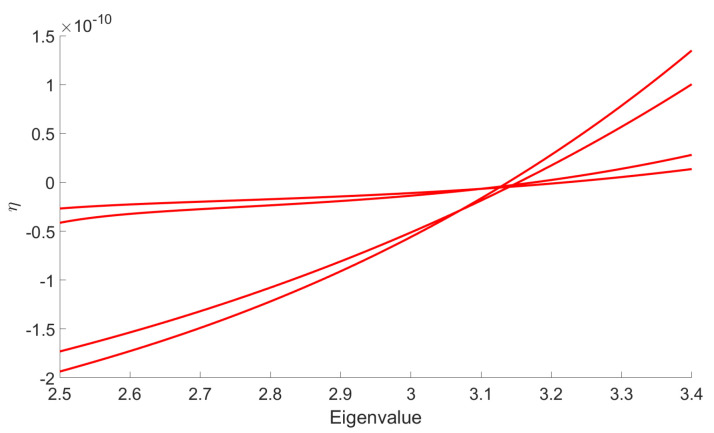
Illustration of the evolution of η, obtained by the LPA’ representation, for eigenvalues between 2.5 and 3.4 in the case of data without signal. We see that for these RG trajectories, the anomalous dimension η remains small. This highlights that there is no significant change when we use the LPA’ representation instead of the LPA one.

## Data Availability

Not Applicable.

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
