# Peer review of "Field Theoretical Approach for Signal Detection in Nearly Continuous Positive Spectra I: Matricial Data"

_entropy, 2021, doi:10.3390/e23091132_

Round 1
Reviewer 1 Report
Dear Editor and Authors,
I believe that this last round of revision does improve the paper and by my current assessment, the paper is sound and lack (in the current version) visible mistakes. As I have mentioned in previous rounds of revision idea of the paper is interesting and valid. Based on those criteria I concur with publication.
However it must be said that, although there has been great improvements, the paper is still convoluted, there is a large amount of math and insufficient focus on results. III A-1, III A-2, and III B-1 The paper is long and even after several readings and revisions I can not name the major result of the paper except for the vague ideas of "RG does what PCA does for a continuous space".
This is likely to render this article not to be sufficiently interesting to the broader readership of Entropy and that should be taken into account when making the final decision.
Author Response
I believe that this last round of revision does improve the paper and by my current assessment, the paper is sound and lack (in the current version) visible mistakes. As I have mentioned in previous rounds of revision idea of the paper is interesting and valid. Based on those criteria I concur with publication.
- We thank the reviewer for agreeing to the publication of our submitted paper.
However it must be said that, although there has been great improvements, the paper is still convoluted, there is a large amount of math and insufficient focus on results. III A-1, III A-2, and III B-1 The paper is long and even after several readings and revisions I can not name the major result of the paper except for the vague ideas of "RG does what PCA does for a continuous space".
- Section A-1 introduces to the reader the general truncation scheme. Section III A-2 provides a detailed derivation of flow equations, namely (32), (35) and (37) in an unconventional context. These equations are essential to obtain Figures 6-7-8 and 10-11-12-13. Finally, Section III B-1 introduces the formalism allowing to reproduce Figures 14-15-16. Note that we provide the technical derivation for anomalous dimension in appendix.
This is likely to render this article not to be sufficiently interesting to the broader readership of Entropy and that should be taken into account when making the final decision.
- We thank once again the referee for his comments, which helps to improve the paper since the first version. Ultimately, this article shows a universal property of continuous spectra in the neighborhood of the MP low, the continuation of this paper, already accepted in the journal shows that it is also true for "tensorial" data and we were able to verify it for all the models running noise.
Reviewer 2 Report
The paper is an interesting contribution to the growing field of renormalization group approaches to the analysis of data. I think the paper has a key problem: the paper is not written to be reproducible.
I say this having worked both in the analysis of data using PCA, and being extremely familiar with the Wetterich-Morris equation and the FRG.
Despite this, I had to keep two brains to follow the paper. One brain was trying to follow the analysis from the point of view of the renormalization group. I was expecting however the other one to be activated towards page 22, when I was assuming some important results about PCA and the choice of Lambda^0. Essentially, as the paper is written is in no way applicable by any practitioner of Machine Learning/PCA/Data Scientists etc. Why should anybody learn any (time consuming !) technique of the exact renormalization group, to do something that I could do with an optimal filter?
It seems to me that the target of this paper is not the audience of an average Entropy reader, but a more targeted audience (which is not even statistical physicists, but just the community of FRG). In this respect, while I truly respect the authors, this work and the analysis performed, the conclusions do not support the publication of this paper as it stands.
In this sense, I wish the authors did pick a very practical example, like the MNIST handwritten digits, added noise, and showed that this method is efficient (in this or that example, does this method suggest how many number of principal components should I pick?). For instance, in that case I would have liked to see a comparison with the minimum description length criterium which is widely used.
I do agree that the source of the approximation is the theoretical embedding.
I do not have any technical concerns about the work otherwise. I think this research direction is worthy and the authors should just consider being a little more practical.
Author Response
The paper is an interesting contribution to the growing field of renormalization group approaches to the analysis of data. I think the paper has a key problem: the paper is not written to be reproducible. I say this having worked both in the analysis of data using PCA, and being extremely familiar with the Wetterich-Morris equation and the FRG.
- Our conclusions in this paper are perfectly reproducible and we will make our codes public as soon as the paper is accepted (We join it to the referee as an additional material in our answer.).
Despite this, I had to keep two brains to follow the paper. One brain was trying to follow the analysis from the point of view of the renormalization group. I was expecting however the other one to be activated towards page 22, when I was assuming some important results about PCA and the choice of $\lambda^0$. Essentially, as the paper is written is in no way applicable by any practitioner of Machine Learning/PCA/Data Scientists etc. Why should anybody learn any (time consuming !) technique of the exact renormalization group, to do something that I could do with an optimal filter?
- The paper is not intended to communicate about an algorithm that is useful in practice. This part will probably be the goal of future investigation. In this work, we continue increasing efforts to connect the important concept of renormalization group (RG) with machine learning and data analysis tools [1-10]. In this investigation, our work follows a fruitful path for the use of RG in a context of applicability to the PCA [11-14]. Namely, the challenging context studied in this series of works tries to address, the problem of signal detection in a continuous spectrum where the distinction between signal and noise appears to be arbitrary with the standard PCA.
- In more details, our aim in this paper is to communicate about a general feature of any continuous spectrum in the neighborhood of the law of MP (rather than a specific aspect regarding some particular problem). Moreover, we have to stress that our conclusions do not only concern MP's law, but seems to be a very general property of models of noise. The same behavior has been observed for tensorial like noises in [14], which follows this paper and have already been published in entropy. Moreover, in our ongoing investigations, we also noticed that it holds for Wigner laws and for a large variety of problems. We now believe that this is a universal property - this universality being the subject of an article in preparation.
- Finally, the use of nonperturbative techniques was justified by the analysis itself. It was a surprise for us to realize that nearly Marchenko-Pastur law, the Gaussian fixed point is unstable (with essentially three directions of instability in the local sector).
[1] Bény, C. (2013). Deep learning and the renormalization group. arXiv preprint arXiv:1301.3124.
[2] Mehta, P., & Schwab, D. J. (2014). An exact mapping between the Variational Renormalization Group and Deep Learning. stat, 1050, 14.
[3] Bény, C. Inferring relevant features: From QFT to PCA. Int. J. Quantum Inf. 2018, 16, 1840012.
[4] Iso, S., Shiba, S., & Yokoo, S. (2018). Scale-invariant feature extraction of neural network and renormalization group flow. Physical review E, 97(5), 053304.
[5] Li, S. H., & Wang, L. (2018). Neural network renormalization group. Physical review letters, 121(26), 260601.
[6] Koch-Janusz, M., & Ringel, Z. (2018). Mutual information, neural networks and the renormalization group. Nature Physics, 14(6), 578-582.
[7] Koch, E.D.M.; Koch, R.D.M.; Cheng, L. Is Deep Learning a Renormalization Group Flow? IEEE Access 2020, 8, 106487–106505.
[8] Chung, J. H., & Kao, Y. J. (2021). Neural Monte Carlo Renormalization Group. Physical Review Research, 3(2), 023230.
[9] Halverson, J.; Maiti, A.; Stoner, K. Neural networks and quantum field theory. Mach. Learn. Sci. Technol. 2021, 2, 035002.
[10] Erbin, H., Lahoche, V., & Samary, D. O. (2021). Nonperturbative renormalization for the neural network-QFT correspondence. arXiv preprint arXiv:2108.01403.
[11] Bradde, S.; Bialek,W. Pca meets rg. J. Stat. Phys. 2017, 167, 462–475.
[12] Lahoche, V.; Samary, D.O.; Tamaazousti, M. Generalized scale behavior and renormalization group for principal component analysis. arXiv 2020, arXiv:2002.10574.
[13] Lahoche, V.; Samary, D.O.; Tamaazousti, M. Field theoretical renormalization group approach for signal detection. arXiv 2020, arXiv:2011.02376.
[14] Lahoche, V.; Samary, D.O.; Tamaazousti, "Field Theoretical Approach for Signal Detection in Nearly Continuous Positive Spectra II: Tensorial Data", Entropy 23(7):795.
It seems to me that the target of this paper is not the audience of an average Entropy reader, but a more targeted audience (which is not even statistical physicists, but just the community of FRG). In this respect, while I truly respect the authors, this work and the analysis performed, the conclusions do not support the publication of this paper as it stands.
- We don't agree with the referee feeling (which we also respect).
We pointed out a universal property of spectra that can be ultimately useful in data analysis. We argue that the audience of an average Entropy reader seems to us more able to appreciate this general message. Indeed, other entropy referees saw great interest as a result of this work by accepting the second part of this paper [Entropy 23(7):795] concerning "tensorial" data.
In this sense, I wish the authors did pick a very practical example, like the MNIST handwritten digits, added noise, and showed that this method is efficient (in this or that example, does this method suggest how many number of principal components should I pick?). For instance, in that case I would have liked to see a comparison with the minimum description length criterium which is widely used.
- These kind of practical applications are not the aim of this paper, which has more theoretical motivations. The use of field theory to study continuous spectra was initiated in [1610.09733]. Our formalism follows the same direction as this reference, but through a formalism more appropriate to general analyzes and not confined to a precise practical problem. Note that for our investigations it was important to study synthetic data for which we have complete control. We voluntary chose to construct a signal from a disturbing of the analytic MP law to keep control on the approximations and on the size of the signal (an image). This was crucial to obtain figures like Fig 11.
- Despite the theoretical interest in connecting RG with ML (and PCA in particular), it is interesting to note that this subject seems to have some practical potential. Indeed, it has already been shown that the task of detecting signal in a continuous spectrum raises quite naturally in different practical applications: in the study of neural activity data [15-17], in biology [18-19] in particular with the study of Single-Cell data [20-21], in genetic data [22] and in financial data [23-24]. Unfortunately, to the best of our knowledge, up to now, there are no publicly available datasets of real data dedicated to this problem. However, we believe that continuing the development of theoretical results on this task, as the proposed one, can encourage practitioner researchers to build and share real datasets containing data that exhibit this continuous spectra behavior. Such datasets could then be used as benchmarks allowing researchers to compete their algorithms against each other. This quite general and well known methodology have been for example exploited for the task of image classification, where large and clean datasets such as MNIST [25] and ImageNet [26] have contributed to a rapid development and even improvement of neural network architectures [27-31].
- Note that we do not address the technical problem of signal recovery. Namely, we are not able at this stage to suggest how many number of principal components should user picks. This problem of signal recovery is out of the scope of this paper.
[15] Tkačik, G., Marre, O., Amodei, D., Schneidman, E., Bialek, W., & Berry, M. J. (2014). Searching for collective behavior in a large network of sensory neurons. PLoS computational biology, 10(1), e1003408.
[16] Meshulam, L., Gauthier, J. L., Brody, C. D., Tank, D. W., & Bialek, W. (2017). Collective behavior of place and non-place neurons in the hippocampal network. Neuron, 96(5), 1178-1191.
[17] Meshulam, L., Gauthier, J. L., Brody, C. D., Tank, D. W., & Bialek, W. (2018). Coarse--graining and hints of scaling in a population of 1000+ neurons. arXiv preprint arXiv:1812.11904.
[18] Agrawal, A., Sarkar, C., Dwivedi, S. K., Dhasmana, N., & Jalan, S. (2014). Quantifying randomness in protein–protein interaction networks of different species: A random matrix approach. Physica A: Statistical Mechanics and its Applications, 404, 359-367.
[19] Korošak, D., & Slak Rupnik, M. (2019). Random Matrix Analysis of Ca2+ Signals in β-Cell Collectives. Frontiers in physiology, 10, 1194.
[20] Aparicio, L., Bordyuh, M., Blumberg, A. J., & Rabadan, R. (2020). A random matrix theory approach to denoise single-cell data. Patterns, 1(3), 100035.
[21] Johnson, E., Kath, W., & Mani, M. (2021). EMBEDR: Distinguishing Signal from Noise in Single-Cell Omics Data. Available at SSRN 3854519.
[22] Xu, Y., Liu, Z., & Yao, J. (2021). ERStruct: An Eigenvalue Ratio Approach to Inferring Population Structure from Sequencing Data. arXiv preprint arXiv:2104.01944.
[23] Laloux, L., Cizeau, P., Bouchaud, J. P., & Potters, M. (1999). Noise dressing of financial correlation matrices. Physical review letters, 83(7), 1467.
[24] Marsili, M. (2002). Dissecting financial markets: sectors and states. Quantitative Finance, 2(4), 297.
[25] LeCun, Y. (1998). The MNIST database of handwritten digits. http://yann. lecun. com/exdb/mnist/.
[26] Russakovsky, O., Deng, J., Su, H., Krause, J., Satheesh, S., Ma, S., ... & Fei-Fei, L. (2015). Imagenet large scale visual recognition challenge. International journal of computer vision, 115(3), 211-252.
[27] Krizhevsky, A., Sutskever, I., & Hinton, G. E. (2012). Imagenet classification with deep convolutional neural networks. Advances in neural information processing systems, 25, 1097-1105.
[28] Szegedy, C., Liu, W., Jia, Y., Sermanet, P., Reed, S., Anguelov, D., ... & Rabinovich, A. (2015). Going deeper with convolutions. In Proceedings of the IEEE conference on computer vision and pattern recognition (pp. 1-9).
[29] Szegedy, C., Vanhoucke, V., Ioffe, S., Shlens, J., & Wojna, Z. (2016). Rethinking the inception architecture for computer vision. In Proceedings of the IEEE conference on computer vision and pattern recognition (pp. 2818-2826).
[30] Szegedy, C., Ioffe, S., Vanhoucke, V., & Alemi, A. A. (2017, February). Inception-v4, inception-resnet and the impact of residual connections on learning. In Thirty-first AAAI conference on artificial intelligence.
[31] He, K., Zhang, X., Ren, S., & Sun, J. (2016). Deep residual learning for image recognition. In Proceedings of the IEEE conference on computer vision and pattern recognition (pp. 770-778)
I do agree that the source of the approximation is the theoretical embedding.
- This is an approximation, surely, but very general, as the assumptions concerning the role of the Z2 symmetry. They can be deduced from an universality argument (going beyond the scope of this conference paper). We could for instance choose to work with a concrete representation of the data at the beginning. It could for example be a set of binary variables $\{s_1,\cdots, s_N\}$, taking entries $s_i\pm1$. A priori we do not know anything about the distribution, apart from the value of the two-point correlations (given by the covariance matrix). We could then make the very natural choice of the maximum entropy distribution (the least structured possible). It is then enough to exploit the properties of the Gaussian integrals to come to the field theory which serves as a starting point (having Z2 symmetry). From there, we could unroll the same arguments and carry out the same numerical experiments on our spectra, and we arrive at the same conclusions. Finally, the universality of the distribution of Marchenko-Pastur would lead us to have to admit that the scope of this field theory goes beyond the initial binary framework, that it forms a class of universality with regard to the distribution itself, without, at this level of descriptions that details about the "real" nature of the random variables play a role. This argument is developed in an upcoming article.
I do not have any technical concerns about the work otherwise. I think this research direction is worthy and the authors should just consider being a little more practical.
- We thank the referee for his compliment, and reiterate that our long-term intention will be to introduce a practical algorithm, but for now we have focused our efforts on deeper theoretical investigations, in particular in the direction of the study of universality.
Reviewer 3 Report
The manuscript presents an approach for signal detection in nearly continuous positive spectra based on some well-known techniques. However the combination of these methods seems to be unique for this particular application. Therefore the authors should be more clear about and better stress the novelty of their work. The following comments are given to further improve the manuscript quality:
- Please remove the references in the abstract.
- Please avoid lumping references, e.g. 2-6 and similar. Instead summarize the main contribution of each referenced paper in a separate sentence and/or cite the most recent and/or relevant one.
- The authors should more clearly explain why they use PCA for their problem and why this method turns out to be more accurate than some others (e.g. SVM).
In overall the contribution of the manuscript is acceptable but it still needs a minor revision.
Author Response
The manuscript presents an approach for signal detection in nearly continuous positive spectra based on some well-known techniques. However the combination of these methods seems to be unique for this particular application. Therefore the authors should be more clear about and better stress the novelty of their work. The following comments are given to further improve the manuscript quality:
- Our conclusions are not specific, but in fact a very general feature of any continuous spectrum in the neighborhood of the law of MP and this is the main message of our paper. Moreover, these conclusions do not only concern MP's law, but seems to be a very general feature of models of noise. The same behavior has been observed for tensorial like noises in [Entropy 23(7):795 -- "Field Theoretical Approach for Signal Detection in Nearly Continuous Positive Spectra II: Tensorial Data"], which follows this paper and have already been published in entropy. Moreover, in our ongoing investigations, we also noticed that it holds for Wigner laws and for a large variety of problems. We now believe that this is a universal property - this universality being the subject of an article in preparation.
Please remove the references in the abstract.
Please avoid lumping references, e.g. 2-6 and similar. Instead summarize the main contribution of each referenced paper in a separate sentence and/or cite the most recent and/or relevant one. The authors should more clearly explain why they use PCA for their problem and why this method turns out to be more accurate than some others (e.g. SVM).
- We voluntary grouped references having non-vanishing intersections as it is standard in the literature. We allowed ourselves to do this grouping mainly for papers having no strong connections with our purpose. Furthermore, we discarded the reference in the abstract.
- The paper is not intended to communicate about an algorithm that is useful in practice. This is a goal that we have, and that we are working on, but not the subject of this article. Our aim in this paper is to communicate about a Universal feature of any continuous spectrum in the vicinity of the MP law (rather than a specific aspect regarding some particular problem).
Round 2
Reviewer 2 Report
The authors have answered to all my questions. I think the paper can be published in the current form.
This manuscript is a resubmission of an earlier submission. The following is a list of the peer review reports and author responses from that submission.
Round 1
Reviewer 1 Report
This work considers interaction-like (non-Gaussian) corrections to multivariate Gaussian distributions and performs a functional RG treatment within the LPA approximation wherein the high energy modes of the Gaussian part are treated as high "momentum" and gradually integrated over to generate an effective theory for the more probable modes. This investigation is motivated by the question of universality and how it may be reflected in PCA. As I understood it, if only a few relevant coupling constants on top of the PCA (or Gaussian Kernel) are enough to describe well the distribution over the low "momentum"--- then there is a signal here, in the sense that one finds a clean measure, based on these couplings, of how distinct or similar are two distributions/noise models.
More specifically the current work adds on top of a previous publication (their reference 20) by accounting for sixth-order interaction terms and performing the RG analysis beyond dimensional consideration.
I found various issues with the current manuscript, due to which I cannot recommend publication.
Presentational issues:
- The paper is extremely wordy, full of suggestive and contradictory remarks. For instance "The fact that only (at least) one subregion can" or "Can we find objective criteria do decide if a continuous signal
contain information or not? There is not a single way to provide an answer for such an issue. " - Due to its general and suggestive writing style, I simply couldn't hold on to a clear refutable message or result.
- Better treatment of related literature is needed. PCA and MP distributions have been studied extensively. Phase transitions in the extraction of a single (a low-rank perturbation) for MP distributions are a classical topic--- yet the authors seem to claim they are "providing
first evidence connecting phase transition and signal detection." - The manuscript is (extremely) full typos. One or two in almost every paragraph.
- More disentanglement of their current results from their Ref. [20] is needed. The current presentation uses either technical or vague terminology on this issue.
Scientific issues
- If the author's approach has predictive power (which I imagine it has), I'd like to see a clear experiment versus theory graph showing what their approach is capable of. In my mind, the current work is too self-absorbed to be of interest to a reader not keenly interested in applying functional RG to data science.
- Their basic model is somewhat ad-hoc, for instance, "non-local" terms such as $\sum_{ij}a_{ij} \phi_i^2 \phi_j^2$ are omitted (for simplicity I guess) although such terms will be relevant for any real data distribution. Can the authors track these matrix terms? If not what are the prospects here in generic scenarios which are non-local?
- With relation to the third point, it feels somewhat artificial to simply apply this approximate method of doing RG, tailored for physical systems with locally, in this far more generic and abstract setting. Simply branding eigenvalues as $p^2$ and performing the standard treatment feels inappropriate to me, or at least the appropriateness of this needs to be discussed clearly.
Notwithstanding my criticism and strong objection to accepting the manuscript in its current form, I do see potential in this approach. My suggestions to the authors are 1. State clearly a well-defined, measurable, and interesting/previously-studied quantity they wish to study. 2. Describe the current state of the art in the field in estimating this quantity. 3. Provide analytical calculations and match their predictions with an experiment.
Author Response
We have grouped all the answers together on the same file.

Reviewer 2 Report
In their work, the authors address the problem of detecting a signal in data of matrix form. In data analysis, this problem is often related to finding a cutoff in the eigenvalue spectrum of a covariance matrix, thereby separating the data into signal and noise. However, for many datasets the observed eigenvalue spectrum is continuous, with no apparent gap, and it is not clear how to set a useful cutoff in such cases. The authors’ attempt to address this problem using techniques from functional renormalisation group theory is timely, ambitious and can potentially provide interesting insights.
The authors’ intention to communicate their results to a broader audience and connect with researchers from the information theory and data analysis communities is visible. Unfortunately, I feel however, that the technical complexity of the methods involved, in particular the functional renormalisation group in the Wetterich-Morris framework, requires a lot more dedication to clear explanation, less focus on details and more focus on concepts and significance of the results. The authors’ expertise in RG techniques, in principle, offers the opportunity for them to bring these methods to a wider audience if they manage to find a pedagogical way to explain their results.
To make the work accessible to non-experts, it might help to address the following points:
- Clear definition of principal component analysis. PCA is certainly a standard technique in data analysis, but since it is so crucial for the main problem addressed in the article, it is only appropriate to state the eigenvector equation. In this context it would also be interesting to answer the basic question, why you perform PCA on the covariance matrix and not on the correlation matrix (normalised form of the covariance matrix) as is also commonly done. I would propose adapting the second paragraph of the introduction for this. Then in this paragraph also the concept of eigenvalue density and as an example the Marcenko-Pastur law could be stated which is currently only provided in Eqn. (41), yet mentioned several times before already.
- The issue with having a continuous spectrum is not explained clearly enough I think. Maybe a figure would be useful here. In their illustration of RG concepts, the authors have indeed used cartoon style Figs. 1-4, which are certainly useful for the audience. Something similar might help to illustrate the cutoff issue.
- Why is the nonperturbative Wetterich-Morris framework needed and why does it improve the quantitative prediction of the signal?
- Compared to reference [19], the authors’ approach focuses on the effective action $\Gamma_k$. They explain in paragraph four of the introduction that this implies the question “what is “noisy” ”, rather than “ what is “information” ”. Why is this an advantage, or is this just another way of approaching the problem?
- A major difficulty with understanding the present paper is that it relies on the authors’ previous work [arXiv:2002.10574]. In its present form, the article is not self-contained, regarding the notation as well as the concepts used.
- Explain already in the introduction how symmetry breaking plays a role in your work. Currently it is only mentioned briefly in the last paragraph of the introduction, but then is a main conclusion in the results section which comes without much context and explanation.
- The results of the work, listed in bullet points at the beginning of Sec. IV Concluding Remarks, are abstract and seem vague. Is there a way to say what this means practically, e.g. what implications the results have for data analysis? For instance, it would help to loop back to the main goal of the paper which is signal detection and provide a concise discussion and assessment.
Other comments and suggestions:
- It might make sense to provide Eqn. (3) as an example right after the concept of the classical action is introduced, before Eqn. (1). This would make the discussion more concrete.
- In the definition of $\zeta_i$ before Eqn. (1) use $\zeta_i\in\mathbf R$. The notation $\zeta_i=]-\infty,+\infty[$ seems unusual.
- In the third paragraph of Sec. II there is an unintelligible sentence “from some physically of mathematical requirements”.
- As far as I can see, no explanation of the term canonical dimension is given. There is a reference to [20] at the end of Sec. IIIA 2., where it is explained, but given the repeated use of this concept later on in the text, I feel that it would be appropriate to provide a self-contained explanation.
- The summation in over ${p_\alpha}$ in Eqns. (13) and (14) is not clear. A look into reference [20] also did not help understanding this. Also, it is not clear to me what the summation index should be here to be consistent with the previous notion: $i$, $\alpha$ or $\mu$?
- The arrows on the flow lines in Figs. 5 and 9 are hardly visible.
- The caption of Fig. 5 is wrong. I think this figure used to contain both the plots from Fig. 5 and 9 and for this reason is still referring to “top” and “bottom”.
- The discussion of the main message of Fig. 6 needs to be better connected to the text.
- Missing axes labels in Figs. 7, 8, 11, 12, 13, 14, 15.
- Axes of Fig. 10 are too small to decipher.
- The authors may want to cite some papers presenting data where continuous eigenvalue spectra appear, e.g. in neuroscience, ethology, financial data, etc.
Author Response

(The authors gave the same response as above.)

Reviewer 3 Report
The article under review analyze the issues of using Principal Component Analysis (PCA) in a continuous spectrum and suggests the use of the Renormalization Groups (RG) as in quantum fields to coarse-grain the effects of short range (or high momentum) interactions. The authors argue that RG can solve the problem of distinguishing the relevant variables in PCA, this is very interesting and sound. However, I have a few questions before recommending publication.
1. The introduction states the Avogadro number as $~10^43$ rather than $~10^24$
2. The authors claim "The connection between PCA and RG can be traced from information theory". Although I agree with this statement, little in the article is made to explain explicitly how this is true. When I think information theory, I think of entropy, is there any references where either PCA or RG equations are derived from purely information frameworks (e.g. by some form of maximum entropy or Bayesian methods)? Is there any calculations of the RG equations (14) from some form of dynamical maximum entropy? It seems clear to me that (5) is the maximum entropy distribution with constraints of normalization, eq (6) and a constraint on the form $\int \Prod d\phi_i \phi^4$.
3. In III-a-3 the MP model is explained. As a reader, I think the article under review comes short of explaining what exactly a MP noise is -- the definition in (41) should be followed by motivation and/or physical meaning -- or what kinds of data one should we expect a MP noise. Why is MP the object of study in the first place?
Minor note, on the paragraph following (41) it reads Rg, I believe the author meant RG for self-consistency.
4. Finally, the authors comment on $\phi^4$ and $\phi^6$ interactions. Considering the goal, of an tensorial PCA, what do the coefficients of these interactions ($g$ in (5) and $u_4$ and $u_6$ in (43)) have any meaning in terms of principal components? Would that mean that PCA should not only consider covariances but also central moments of order 3,4,5, and 6?
I have other concerns with the article under review:
Many aesthetics problems appear (e.g. the quotation marks are two 'closed quotations' everywhere except the references). Also, in order to guarantee reproducibility of the research, I would strongly suggest the code that generate the numerical calculation of most figures to be available -- e.g. on GitHub. Also, in order to facilitate review work and feedback the authors could enumerate the lines in the manuscript (using the Latex package lineno for example, which is already given in the MDPI template)
Author Response

(The authors gave the same response as above.)

Reviewer 4 Report
The manuscript focuses on signal detection and exhibition of experimental evidence in favor of a connection between symmetry breaking and the existence of an intrinsic detection threshold. In my opinion, the manuscript can be accepted for publication only after moderate revision according to the given recommendations. Some remarks are given in the attached document as well.
Minor remarks
The scientific manuscript should be written in the third-person singular. The first-person plural should be omitted from the manuscript.
Major remarks
Avoid lumping the references. Each study should be discussed separately.
The reference list should be reduced.
The conclusion should be retyped and reduced. Only the main conclusions should be mentioned in this section. The cited references should be excluded from the manuscript.

Author Response

(The authors gave the same response as above.)

Round 2
Reviewer 1 Report
I have gone over the author's reply and the manuscript. The authors haven't made extensive changes to the manuscript. In particular, they didn't contrast their results with the current state of the art and many presentation issues are still problematic (typos in particular). I leave my overall assessment as is.
Author Response
Dear referees, dear editor,
Please find here the detailed answer to the referee comments.
1 Answer to referee 1
I have gone over the author’s reply and the manuscript. The authors haven’t made extensive changes to the manuscript. In particular, they didn’t contrast their results with the current state of the art and many presentation issues are still problematic (typos in particular). I leave my overall assessment as is.
• We have made a very large number of corrections to the manuscript, and we have taken into account the referee’s remark on the state of the art. In particular:
– We have added a paragraph in the introduction which recalls the standard paradigm of PCA, in particular with regard to the aspects related to the phase transition associated with the detection.
– We have also mentioned several references, related to our work. The use of field theory to study continuous spectra was initiated in [1610.09733]. Physical interpretations of the field theory in this con- text of data analysis have been made in [1612.08935, 1812.11904]. For instance, they interpret the "vacuum" in terms of average activity of neurons. Our formalism follows the same direction as these references, and then is not as stripped of the literature, as commented the referee in its remark.

Reviewer 2 Report
I would like to thank the authors for their efforts in addressing my questions and improvement of the manuscript. Unfortunately, I still do not think that the article is suitable for publication, for the following reasons: 1. The article is still written without the attention to detail that I would expect from a scientific publication. Let me provide just one of many examples: There is this sentence in the paragraph before Eq. (1), where I had pointed out an issue with the notation in my previous report (which the authors have fixed):"It can be for example a discrete variable, as for the Ising model [41], where ζi = ±1, a real number ζi ∈ R."
The authors talk about discrete numbers and real numbers at the same time and things end up being confusing. Maybe now there is just a word missing or something, but I would have expected this sentence to come back fully correct, since all that is needed is a little bit of attention when editing things.
2. The main question and message of the article are still too convoluted for it to be accessible to a wider audience. It is hard for me to provide sufficient suggestions here, since I think there are simply too many basic points that need to be addressed to achieve clarity.
3. I am still very uncomfortable with how much this work relies on the article [arXiv:2002.10574], which as far as I can see has not been peer reviewed. This makes it almost impossible to provide solid, reliable judgement of the present work.
Author Response
Dear referees, dear editor,
Please find here the detailed answer to the referee comments.
2 Answer to referee 2
1. The article is still written without the attention to detail that I would expect from a scientific publication. Let me provide just one of many examples: There is this sentence in the paragraph before Eq. (1), where I had pointed out an issue with the notation in my previous report (which the authors have fixed): "It can be for example a discrete variable, as for the Ising model [41], where ξi = ±1, a real number..."
• We just wanted to list in this sentence some examples of what ξi might be, a discrete variable, or a real number, or whatever, and the sentence should have ended with "..." instead of a "." We have removed this sentence in the new version.
2. The main question and message of the article are still too convoluted for it to be accessible to a wider audience. It is hard for me to provide sufficient suggestions here, since I think there are simply too many basic points that need to be addressed to achieve clarity.
• We strongly improved the clarity and the presentation in the revised version. Moreover, in regard to our results, they are announced in the introduction and enumerated in the conclusion. To put in a nutshell, we connect the presence of a signal with a lack of symmetry restoration for a compact region in the vicinity of the Gaussian point, as Figures 11 and 12 showed.
3. I am still very uncomfortable with how much this work relies on the article [arXiv:2002.10574], which as far as I can see has not been peer reviewed. This makes it almost impossible to provide solid, reliable judgement of the present work.
• We mentioned several references, of which our work can be seen as direct continuations with respect to the initial formulation issue. The theoretical
2
entropy-1229316
formalism was introduced in [1610.09733] where the problem posed by continuous spectra was discussed. In [arXiv:1612.08935, 1812.11904] for instance, the "vacuum" with which we work is interpreted in terms of average activity of neurons.
• We would like to mention that, the paper [arXiv:2002.10574] has been submitted for over a year to a journal, and unfortunately we are still waiting for the first referee comments... Also, the companion paper of this article [arXiv:2012.07050] has recently been accepted in Entropy journal (Entropy (2021), 23, 795.). Finally, our results are completely independent of this reference [2002.10574] which is restricted to the study of power-counting. In the proposed paper, we have gone far beyond by the observation of the existence of a phase transition controlled by the intensity of the signal in a data.

Reviewer 3 Report
Unfortunately, the authors did a subpar work in revising the manuscript. It seems as if they did not take the necessary time to prepare it to the next stage of revisions. I know sometimes deadlines can be stressful but I believe the authors deserve to take the necessary time to fully revise the manuscript -- that can come either by requesting an extension to the deadlines or by withdrawing and submitting again. But, as of now, this manuscript is not suitable for publication.
Several aesthetics problem still occur. The authors say in their reply that they were solved but I could still find many -- some examples are explained in the attached file -- including some even in the specific example of quotation marks. That includes, but is not restricted to, typos but goes beyond to the point that the manuscript is hard to read with so many incomplete sentences lacking verbs or prepositions.
Their answers to my report also leaves much to be desired.
On question 2 - "Fisher information metric in that context (which is nothing but an infinitesimal version of the Kullback-Leibler Divergence)"
I disagree -- a sentence similar to this is present in the manuscript -- The information metric does match the KL divergence in the sense for distributions in the exponential family the infinitesimal distances equate to the relative entropy in the second order. The mathematics is more subtle than the authors' statement indicates. Moreover, an explanation on information geometry is necessary.
On the same question - agreement to my question was given, but no explanation of my question 2 is given in the manuscript.
On question 3 - The sentence before the MP law lead me to more confusion, what experiments are the authors talking about? I may have missed this in the first round of reviews, but are they using the word 'experiment' to mean their numerical simulation with MP noise?
Regretfully the suggestion to number the lines (or use the format), thus helping the review work, was ignored as was my suggestion to make the code available. The authors justify their fail to provide reproducibility of research by stating "we think that it is easily reproducible from theequations given in the document." which I strongly disagree. Even if the equations were easy (which I do not believe they are) the work necessary when doing numerical analysis always include a choice of appropriate library and there is always some methods who do not work. Making the code available is, therefore, of uttermost importance. I do understand that some authors might be concern to publish their code before publication, but if they have said so I would ask to send it as supplemental material to reviewers.

Author Response
Dear referees, dear editor,
Please find here the detailed answer to the referee comments.
3 Answer to referee 3
Several aesthetics problem still occur. The authors say in their reply that they were solved, but I could still find many – some examples are explained in the attached file – including some even in the specific example of quotation marks. That includes, but is not restricted to, typos but goes beyond to the point that the manuscript is hard to read with so many incomplete sentences lacking verbs or prepositions.
• We thank the referee for its comment, and we strongly improved the presentation in accordance with its advises in the current version.
Their answers to my report also leaves much to be desired. On question 2 - "Fisher information metric in that context (which is nothing but an infinitesimal version of the Kullback-Leibler Divergence)" I disagree – a sentence similar to this is present in the manuscript – The information metric does match the KL divergence in the sense for distributions in the exponential family the infinitesimal distances equate to the relative entropy in the second order. The mathematics is more subtle than the authors’ statement indicates. Moreover, an explanation on information geometry is necessary.
• Concerning the KL divergence, it seems that the word “version” can be confusing. Indeed, finally, we think that the referee and us are referring to the same elementary result. Since the value at 1, and the first derivative of KL divergence vanishes, the first non-zero term comes from the second derivative, which matches with the Fisher metric. We have removed this sentence from the Manuscript to avoid confusion.
On the same question - agreement to my question was given, but no explanation of my question 2 is given in the manuscript.
3
entropy-1229316
• In regard to the question of relation between the RG equations and some form of dynamical maximum entropy? These questions have been investigated, see for instance "Exact Renormalization Groups As a Form of Entropic Dynamics" by Pedro Pessoa and Ariel Caticha (Arxiv: 1712.02267, also published in Entropy), establishing a clear link between the RG and information theory. These aspects could be very interesting for future, it could in particular be interesting to build a formalism centered on the notion of entropy. We added the reference (ref [40]) in the revised manuscript, and a comment in the conclusion.
On question 3 - The sentence before the MP law lead me to more confusion, what experiments are the authors talking about? I may have missed this in the first round of reviews, but are they using the word ’experiment’ to mean their numerical simulation with MP noise?
• Yes, experiment mean numerical simulation. In the current version, we replaced this term with "numerical simulations" or "numerical investigations", to be clearer.
Regretfully, the suggestion to number the lines (or use the format), thus helping the review work, was ignored as was my suggestion to make the code available. The authors justify their fail to provide reproducibility of research by stating "we think that it is easily reproducible from the equations given in the document." which I strongly disagree. Even if the equations were easy (which I do not believe they are) the work necessary when doing numerical analysis always include a choice of appropriate library and there is always some methods who do not work. Making the code available is, therefore, of uttermost importance. I do understand that some authors might be concern to publish their code before publication, but if they have said, so, I would ask to send it as supplemental material to reviewers.
• We understand the point of view of the referee, and we added the line numbers in the paper. We also attached as a supplementary material our Matlab code. The provided code enables the referee to reproduce the main figures of the document, namely Figures 6, 9, 10 and 11.

Round 3
Reviewer 2 Report
No Answer
Reviewer 3 Report
I am happy to see that this version of the manuscript does improve significantly on the past work. I still, however, have many further questions and suggestions before recommending publication.
Line 10/11 - Very large data sets or highly correlated data sets?
Line 18 - "experimental". The authors indicated in the previous email that they were to change this terminology to numerical investigations.
Line 26 - what do the authors mean by 'other methods'? Classical Mechanics?
Line 36 - I highly disagree with the statement "Data science ... application for these physical concepts". If they refer to RG, is important to notice that RG in data analysis is not conventional. If they mean that all data science is physical this is a highly controversial statement, not all of data science is based on direct physical information. Maybe the authors meant information theory instead of physics.
Line 38 - Does "extract information" here mean find relevant features?
Line 41/42 - covariance matrix is defined as the second central moment as in any textbook, it is later identified as X^T X
Line 42 - "2-point correlations" This is the result of an analgy to QFT, that does not necessarily is general in Data science
Line 59 - In information geometry the distances are given by the information metric, not by KL, see the past round of reviews. The authors understand that this similarity is only up to second order and this should be explained in the text. An explanation of information geometry (with references) is also in order.
Line 71 (and beyond) - A definition of the terms IR and UV is necessary, not only to desribee the acronyms, but because reader of entropy are not necessarily from QFT/Particle physics backgrounds and these terms are not used literally.
Line 84/92 - give reference for the "one-spiked matrix model" and define \lambda, this way I can't follow this paragraph.
Line 123 - Do the authors mean a functional integral for continuous degrees of freedom?
Line 134 - I believe 'a la Wilson' could be replaced by 'as described by Wilson' with references.
Eq 3, 6, 7 - What are these integrals performed on? are they functional integrals in \phi?
Lines 220/221 - Can this statement "derived from maximum entropy" be followed by references?
Eq13 - apparently p_\alpha are defined in real values, can the Kronecker delta be replaced by a Dirac delta?
Is it possible that the calculations on III A 1, III A2 and III B be substituted by their relevant results? and if the calculation is necessary, be moved to an appendix?
On a minor note, the references are NOT in order which is easy to fix in latex.
The paper is, still, cumbersome and although the authors improved their presentation I still don't think the layout is helpful to the readers. Albeit the interest in the submitted manuscript, extensive revisions are necessary.